# Design, structure and plasma binding of ancestral β-CoV scaffold antigens

David Hueting[1,2,9], Karen Schriever [1,2,9], Rui Sun[3], Stelios Vlachiotis[3], Fanglei Zuo [3], Likun Du[3], Helena Persson[2,4], Camilla Hofström[2,4], Mats Ohlin [4,5], Karin Walldén[6], Marcus Buggert [7], Lennart Hammarström[3], Harold Marcotte [3], Qiang Pan-Hammarström [3], Juni Andréll [6,8] ✉ & Per-Olof Syrén [1,2] ✉

We report the application of ancestral sequence reconstruction on coronavirus spike protein, resulting in stable and highly soluble ancestral scaffold antigens (AnSAs). The AnSAs interact with plasma of patients recovered from COVID-19 but do not bind to the human angiotensin-converting enzyme 2 (ACE2) receptor. Cryo-EM analysis of the AnSAs yield high resolution structures (2.6–2.8 Å) indicating a closed pre-fusion conformation in which all three receptor-binding domains (RBDs) are facing downwards. The structures reveal an intricate hydrogen-bonding network mediated by well-resolved loops, both within and across monomers, tethering the N-terminal domain and RBD together. We show that AnSA-5 can induce and boost a broad-spectrum immune response against the wild-type RBD as well as circulating variants of concern in an immune organoid model derived from tonsils. Finally, we highlight how AnSAs are potent scaffolds by replacing the ancestral RBD with the wild-type sequence, which restores ACE2 binding and increases the interaction with convalescent plasma.

The pandemic caused by severe acute respiratory syndrome coronavirus 2[1] (SARS-CoV-2) has had devastating consequences on global health and economy[2]. Despite the success of vaccination campaigns[3], emerging variants[4–6] are of concern and novel viruses with the potential to drive future pandemics are circulating in nature[7,8]. Phylodynamic studies have shown that SARS-CoV-2 acquires, on average, one substitution every 11th day[9]. Several SARS-CoV-2 variants display enhanced infectiousness and/or increased mortality rates as well as evasion from immunity from vaccination or previous infection[6]. For the omicron variant BNT162b2 (Pfizer-BioNTech) vaccine effectiveness was reduced from about 90% to 70%[10]. Despite the rapid development

of several vaccines against SARS-CoV-2, the lack of comprehensive protection against infection with emerging variants therefore continues to present a risk to global health care systems.

Moreover, beta-coronaviruses have previously caused epidemics (severe acute respiratory syndrome (SARS), Middle East respiratory syndrome (MERS)) and searches in available sequence data revealed many previously unknown and potentially pathogenic coronaviruses[8]. For instance, an ACE2-dependent sarbecovirus able to enter human cells has been identified in Russian bats[11], highlighting reservoirs of viruses in wild animals that have the potential to drive future pandemics. Platform antigens that could elicit cross-reactive immune

[1]School of Engineering Sciences in Chemistry, Biotechnology and Health, Department of Fibre and Polymer Technology, KTH Royal Institute of Technology, Stockholm, Sweden. [2]School of Engineering Sciences in Chemistry, Biotechnology and Health, Science for Life Laboratory, KTH Royal Institute of Technology, Stockholm, Sweden. [3]Division of Immunology, Department of Medical Biochemistry and Biophysics, Karolinska Institutet, Stockholm, Sweden. [4]Drug Discovery and Development Platform, Science for Life Laboratory, Solna, Sweden. [5]Department of Immunotechnology, Lund University, Lund, Sweden. [6]Department of Biochemistry and Biophysics, Science for Life Laboratory, Stockholm University, Stockholm, Sweden. [7]Center for Infectious Disease, Department of Medicine Huddinge, Karolinska Institutet, Stockholm, Sweden. [8]Department of Medical Biochemistry and Biophysics, Karolinska Institutet, Stockholm, Sweden. [9]These authors contributed equally: David Hueting, Karen Schriever. ✉e-mail: juni.andrell@scilifelab.se; per-olof.syren@biotech.kth.se

responses against different coronaviruses, and that can be quickly adapted to novel virus strains, or sub-variants, would therefore be a powerful tool in combating the most recent and future coronavirus pandemics.

Many sequence differences between different SARS-CoV-2 variants and variant sublineages have been mapped to the spike protein (S protein), a homotrimeric glycosylated membrane fusion protein. S protein initiates cell entry by associating with the major host cell receptor, the human angiotensin-converting enzyme 2 (ACE2), via its receptor-binding domain (RBD, residues 319–541 in SARS-CoV-2)[12]. The S protein binds the ACE2 receptor in the so-called RBD-up conformation[12]. The S protein is also the major antigen targeted by vaccines and therapeutic antibodies. However, low production titres and low protein stability are obstacles for developing S- and other protein-based vaccines that can be globally distributed at room temperature. Additionally, low antigen stability can be associated with a reduced immunological response[13].

The state of the art in antigen engineering lies in structure-guided single amino acid substitutions (schematically shown in Fig. 1)[14–17], which has resulted among others in proline S protein variants, including the S2P[12,18] and HexaPro[14] constructs with retained pre-fusion conformations, increased expression yields and thermal stability. Notably, some of the approved vaccines (such as the Pfizer and Moderna mRNA vaccine constructs, Janssen adenovector vaccine and Novavax protein subunit vaccine) are based on the S2P construct[19], a modification without which production levels are negligible (the latter two vaccines also have a mutated furin cleavage site).

Existing sequence space has enabled study of viral emergence, spread and epidemiological dynamics of coronaviruses[20]. We wondered if the wealth of homologous S protein sequences available in databases could also be used for antigen engineering. Ancestral sequence reconstruction (ASR) is a bioinformatics technique that infers nucleotide or amino acid sequences of ancestral nodes in a phylogenetic tree, based on the entire underlying sequence information and topology. Many previously described reconstructed ancestral proteins have been reported to exhibit increased solubility and stability compared to the corresponding extant proteins[21,22]. We were therefore interested in applying this technique to the full-length SARS-CoV-2 S protein (Wuhan wt, UniProt P0DTC2) to obtain stable protein variants without the need for a structural template. Hence, we constructed ancestral scaffold antigens (AnSAs) of the betacoronavirus S protein family (Fig. 1). Several AnSAs were obtained in high yield and purity without the need for rational mutagenesis and exhibited favourable stability and solubility. Additionally, the AnSAs interacted with plasma samples of recovered COVID-19 patients, but not the ACE2 receptor. Cryogenic electron microscopy (cryo-EM) structures with high local resolution indicate how an intricate network of hydrogen bonds between the RBD and N-terminal domain (NTD) maintains the AnSAs in a closed pre-fusion conformation. We further show how an AnSA was able to induce or boost a specific immune response in an immune organoid model derived from tonsils of infected and/or vaccinated donors (referred to as "tonsil organoids" from here on). Finally, we engineered the AnSAs to host the RBD of the wild-type (wt) SARS-CoV-2 S protein, which restored ACE2 binding and resulted in antigens that were bound and neutralized by antibodies in patient sera.

## Results
### Reconstruction of ancestral S proteins
A basic Local Alignment Search Tool (BLAST) search for SARS-CoV-2 S protein homologues was performed in the end of 2020, identifying S

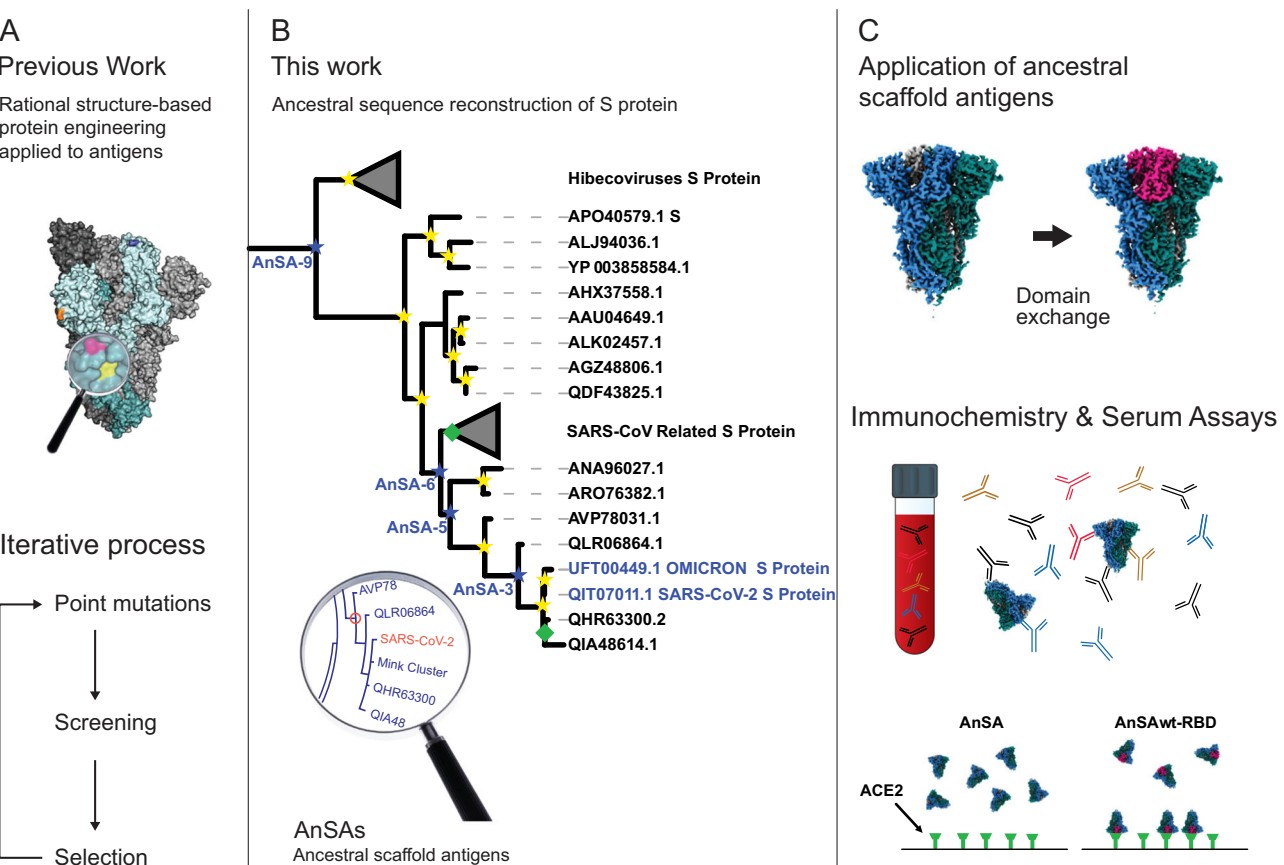

**Fig. 1 | Previous and present work. A–C** Phylogenetic tree highlighting S-protein of SARS-CoV-2 and ancestors with Omicron variant shown for reference (yellow star 100 bootstrap, green diamond >90 bootstrap, blue star AnSA nodes experimentally characterized). Full tree used for ancestral sequence reconstruction is shown in Supplementary Fig. 1.

protein sequences from numerous betacoronaviruses (*Sarbecovirus*, *Merbecovirus*, *Nobecovirus* and *Hibecovirus* lineages), out of which 60 were used to construct a maximum likelihood phylogenetic tree (Fig. 1, full tree in Supplementary Fig. 1). All except five of these sequences were available at the onset of the COVID-19 pandemic. Building a tree exclusively using those sequences that were available as of 1/1/2020 yielded highly similar topology and ancestral sequences (Supplementary Fig. 1, box).

The S protein appears to be a suitable candidate to construct a phylogenetic tree, since the above mentioned betacoronavirus lineages were clearly distinguishable. Nevertheless, it is well-established that recombination events are common in the coronavirus spike protein, in particular in proximity of the receptor binding domain (RBD) and N-terminal domain[20,23,24]. For instance, the RBD of bat SARS-like coronavirus ZC45 (GenBank AVP78031.1) S protein, which is located close to extant SARS-CoV-2 S protein in the phylogenetic tree, shows recombination with the RBD from non ACE-2 binding β-coronaviruses (HKU3 related)[25]. We decided to exclude recombination events as a factor when building the phylogenetic tree, to verify the ASR technique as a straightforward immunogen engineering approach, without extensive analysis of ancestry or origin of the virus. As such, the full ancestral sequences generated herein might not capture the entire evolutionary history of the different virus lineages, yet are hypothesized to capture stabilizing mutations.

We chose to experimentally study several nodes upstream of SARS-CoV-2 S protein, namely nodes 3, 5, 6 and 9 (Supplementary Fig. 1). Nodes 3 and 5 were selected because they are closely related to the extant S protein sequence of SARS-CoV-2. Node 6 was chosen as the node closest to SARS-CoV-2 S protein that precedes both the branches of SARS-CoV-2 and a group of sequences closely related to SARS-CoV. Node 9 in turn represents the closest node that precedes S proteins of *Sarbecoviruses* and non-*Sarbecoviruses*, namely *Hibecoviruses*. Therefore, we were also interested in experimentally studying node 9. The ancestral sequences representing these nodes were referred to as AnSA-3, AnSA-5, AnSA-6 and AnSA-9 and share 83, 79, 80 and 55 % sequence identity with extant SARS-CoV-2 S protein (Wuhan wt), respectively (Supplementary Fig. 2). While the characteristic furin cleavage site of SARS-CoV-2 S protein ($Q_{677}$TNSPRRAR-SV$_{687}$) is lacking in the investigated AnSAs, a single Arginine trypsin cleavage site remains at the S1/S2 interface in AnSA-3, -5 and -6 (but not -9) (Supplementary Fig. 2). The AnSAs do not contain the proline mutations that were designed into the wt sequence to generate the S2P and HexaPro variants.

## AnSA expression and cryo-EM structures

Previous studies have reported that expression yields of the wt SARS-CoV-2 spike protein and its 2P-stabilized variant are low. We therefore decided to use HexaPro[14]—one of the best expressible and thermostable variants of the SARS-CoV-2 S protein known and obtained from structure-based mutagenesis—as a reference. AnSA-3 was expressible, albeit in yields that were lower compared to HexaPro (Supplementary Fig. 3A). The expression of the oldest of the ancestors considered in this study, AnSA-9, was not detectable on an SDS-PAGE gel. In contrast, pure AnSA-5 and AnSA-6 could be obtained from Expi293F supernatants in yields that are comparable or slightly favourable to HexaPro. At the optimal time (3 days) between transfection and harvesting, the yield of AnSAs is ca. 60% higher compared to HexaPro (Supplementary Fig. 3B, C). To test aggregation propensity, purified HexaPro and AnSA proteins were analysed by dynamic light scattering (DLS) (Supplementary Fig. 3D–G). HexaPro, AnSA-5 and AnSA-6 all showed a major peak at a hydrodynamic radius of 30–40 nm, likely representing the correctly folded protein trimer. All three samples also showed a smaller broad peak at a higher hydrodynamic radius (ca. 420–670 nm) which likely represents a collection of different aggregate species. HexaPro exhibited an additional peak in the mid-range size at

approximately 150 nm, which may represent a fraction of folded HexaPro molecules populating a different extended conformation or a smaller type of aggregate. Interestingly, the area-ratio of aggregate to soluble peak was lower in AnSA-5/6 samples (0.36 and 0.28, respectively) compared to HexaPro (0.55) and all three proteins had similar polydispersity indices (PIs) of ca 0.5 (a PI < 0.25 indicates a monodisperse protein sample), in line with their similar expression yields. In contrast, AnSA-3 showed a considerable aggregate peak as well as a continuum of several different overlapping peaks, indicating the presence of a variety of conformations and types of aggregates (Supplementary Fig. 3E). In line with this observation, the PI for this protein sample was ca. 0.94, quantitatively reflecting the inhomogeneity of the sample. This observation may explain the lower purification yields obtained for this variant.

The structural integrity of AnSA-5 and AnSA-6 was confirmed by cryo-EM. One dataset of each protein was collected in one session and readily yielded 3D reconstructions at an overall resolution of 2.59 and 2.77 Å for AnSA-5 and AnSA-6, respectively (Supplementary Table 1 and Supplementary Fig. 4). The overall structures of AnSA-5 and AnSA-6 are similar to the S protein of SARS-CoV-2 (root-mean-square deviation (RMSD) of ca. 1.0 Å per monomer when superposed on PDB 6VXX[26] for 522 (AnSA-5) and 529 (AnSA-6) pruned atom pairs, respectively), indicating a conserved overall structure that is maintained in the ancestral S proteins. Both reconstructions show the ancestral spike proteins in tightly locked pre-fusion conformations with all three RBDs down (Fig. 2 and Supplementary Fig. 5), which was the sole conformation that could be mined from the cryo-EM data. In contrast, many previous structural studies of SARS-CoV-2 S protein reported the presence of multiple different conformations of the RBD domain[26,27].

A high local resolution extends to the loop regions, especially in AnSA-5 (Supplementary Fig. 4), showing that helix/loop motifs in the SD2 domain (yellow loop in Fig. 2) and in the S2 domain (orange loop in Fig. 2) pack tightly against the RBD and NTD domains. This clear resolution of loops that reside at the junction between the SD2, RBD and NTD domains within and across monomers indicates that the domain interfaces are rather rigid in AnSA-5/6. The same loops were well resolved in a tightly locked version of the closed pre-fusion structure of the SARS-CoV-2 S protein that had been obtained by disulphide engineering[16].

While conventional pre-fusion locking strategies either target the core of the protein (cavity-filling mutations) or introduce physical locks into the structure (disulphide bonds, prolines), the domain interface in AnSAs appear to be stabilized by an altered hydrogen-bonding network between the domains. In fact, the majority of the mutations between AnSAs and the Wuhan wt sequence are located on the surface of the trimer (Supplementary Fig. 6) and many of them allow for the formation of hydrogen bonds (Supplementary Datasets 1 and 2). For instance, S284 (A292 in SARS-CoV-2 S protein) creates a novel hydrogen bond between its sidechain (NTD) to the backbone of F310 that is located immediately adjacent to the RBD domain (Fig. 2A, bottom left). This NTD-RBD interface is further connected to the SD2 domain through a hydrogen bond between T604 (SD2) and the backbone of Q285 (NTD). Additionally, the sidechain of Q602 (SD2) forms a hydrogen bond to the backbone of C512 (RBD), thus forming a stabilized tri-loop interface between the NTD-, RBD and SD2 domain of one monomer (Fig. 2A, bottom left).

Besides altering inter-domain interfaces, several surface mutations in AnSA-5 are involved in hydrogen bonding across monomers. For example, the RBD in monomer A and the NTD in monomer B are held together by a hydrogen bond bridging S386 and E490 in the RBD (N394 and E516 in SARS-CoV-2 S protein) and a salt bridge connecting E490 (RBD) and K226 in the NTD domain (D228 in SARS-CoV-2 S protein) (Fig. 2A, top right). In monomers A and C, K226 may form an additional salt bridge to D46 in the NTD domain (K41 in SARS-CoV-S S protein). Similarly, the protomer interface between RBD (monomer A),

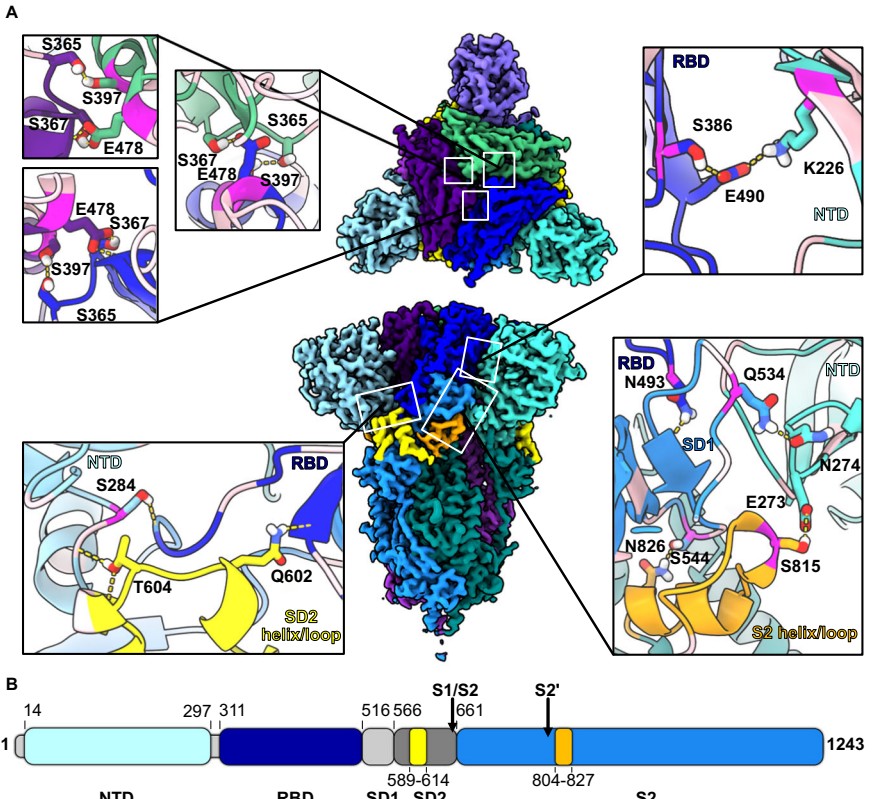

**Fig. 2 | Cryo-EM structure of AnSA-5. A** Top view (top) and side view (bottom) of AnSA-5 electron density map generated by cryo-EM. Each one monomer is coloured in blue (monomer A), green (monomer B) and purple (monomer C) with the NTD domains highlighted in lighter shades and the RBD domains highlighted in darker shades. A helix/loop motif in the apex of the SD2 domain is coloured yellow and an adjacent helix/loop motif on the S2 domain is coloured orange. Interdomain hydrogen bond networks are shown in boxes: top left RBD-RBD interfaces; top right RBD (monomer A)-NTD (monomer B) interface; bottom left NTD-RBD-SD2 interface; bottom right NTD(monomer B)-RBD(monomer A)-S2 (monomer B). Side chains of residues that form hydrogen bonds at these interfaces are shown in stick representation and are labelled, hydrogen bonds are indicated as dashed yellow lines. The $C_\alpha$ atoms of residues that are mutated with respect to SARS-CoV-2 S protein are indicated in pink (dark pink if they have different sidechain properties and enable a novel hydrogen bond, Supplementary Dataset 1). A glycosylation on residue 274 is omitted for clarity. **B** Domain overview as bar graph with domain boundaries indicated above the bar. Colours are the same as in (**A**) and correspond to domain colouring in monomer A. The SD1 and SD2 domains are coloured light and dark grey (grey omitted in (**A**) for clarity).

NTD (monomer B) and S2 helix/loop (monomer B) is stabilized by hydrogen bonds between sidechains of S815 (S2) and E273 (NTD) (A854 and E281 in SARS-CoV-2 S protein) as well as N274 (NTD) and Q534 (RBD) (N282 and L560 in SARS-CoV-2 S protein) (Fig. 2A, bottom right). All AnSA sidechains discussed above that confer interactions not existing in extant protein are conserved between AnSA-5 and AnSA-6 (S284/S286, S386/S388, K226/K228, D46/D46, S815/S834 and Q534/Q553 in AnSA-5/6, respectively). Likewise, mutations in AnSA-5-RBD such as S397 and E478 (corresponding to D405 and G504 in SARS-CoV-2 wt-RBD) allow for formation of inter-RBD hydrogen bonds (Fig. 2A, top left) in addition to conserved ones. In contrast, the RBD trimer interface of the closed wt structure (S2P variant, PDB-ID 6VXX[26]) is more open. Such altered inter-domain hydrogen bonds likely affect the packing and flexibility at the domain interfaces.

### Suitability of AnSAs as antigens

A protein-based vaccine needs to be stable in order to meet logistic challenges related to storage and distribution. The high local resolution of loops in AnSA-5 and -6 (Supplementary Fig. 4) indicates that these proteins are rather rigid overall, which implies a certain inherent protein stability. We experimentally verified thermal stability of AnSAs compared to HexaPro using differential scanning fluorimetry (DSF). HexaPro showed two separate $T_m$ values at ca. 47 and 63 °C (yellow line in Supplementary Fig. 7A), similar to the temperatures originally reported for this construct (ca. 51 and 67 °C)[14] that have been suggested to represent unfolding of the RBD domain and trimer,

respectively[28]. While the melting profile of AnSA-5 was similar to that of HexaPro (blue line in Supplementary Fig. 7A), its overall thermo-stability was higher with $T_m$ values of 50 and 67 °C, representing an increase of 3 and 4 °C compared to HexaPro, which itself was reported to exhibit a ca. 5 °C increase in the first $T_m$ compared to the S2P variant[14]. In AnSA-6 (red line in Supplementary Fig. 7A) the intensity of the two peaks was inverted and associated with $T_m$ values of 52 and 61.4 °C, representing a difference of 5 °C and −1.6 °C compared to HexaPro, respectively.

The thermal stability of AnSAs in the presence of denaturants was measured with label-free nanoDSF, a method that measures changes in autofluorescence of aromatic side chains due to protein unfolding. In nanoDSF experiments, HexaPro exhibited a clear unfolding trace with a major peak at 63.0 °C (yellow trace in Supplementary Fig. 7B), similar to the second $T_m$ in DSF. The first $T_m$ observed in DSF is not visible in the nanoDSF trace. While HexaPro appeared to unfold in a concerted manner, the unfolding of AnSA-5 and -6 as measured by nanoDSF was more complex (blue and red traces in Supplementary Fig. 7B), involving multiple peaks which may represent partial and independent unfolding of different constituent domains. These partial unfolding events are likely not captured by DSF, since the SYPRO Orange dye binds to fully exposed hydrophobic unfolded regions. Adding 2 M urea during thermal unfolding shifted the melting temperatures of HexaPro to lower temperature while maintaining the unfolding pattern (yellow trace in Supplementary Fig. 7C), whereas the unfolding patterns of AnSA-5 and

-6 changed more drastically (blue and red traces in Supplementary Fig. 7C).

Urea is suggested to denature proteins by intercepting hydrogen bonds[29], reinforcing the observation that AnSA stability is dependent on hydrogen bonding networks to maintain the closed pre-fusion state. When melting the proteins in presence of 2 M guanidine hydrochloride (GnHCl), the melting temperatures of all three proteins converged to a similar range between ca. 60 and 70 °C (Supplementary Fig. 7D). It has been reported that GnHCl induces proteins to assume a molten globule-like state[30], in which secondary structure elements are retained but internal packing is lost. The results are in accordance with the observation that the majority of the mutations in AnSAs are distributed on the protein surface (Supplementary Fig. 6). Backscattering data recorded concomitantly to the thermal denaturation data showed that no larger aggregates are visible at any of these conditions (Supplementary Fig. 7E–H).

Furthermore, protein-based vaccines would benefit from both prolonged shelf life and bioavailability of the antigen, as mediated by increased protein stability. It is conceivable, that the latter would also apply to antigens produced in vivo by mRNA/DNA vaccines. We therefore characterized protein shelf-life stability at different temperatures (cold, room temperature and body temperature) over an extended time period. DLS analysis of the same AnSA and HexaPro samples taken at different time points over the course of a month showed that the relative amount of folded protein remained stable at

room temperature and 37 °C for ANSA-5 and -6. After one-month storage at 37 °C, the amount of aggregates in AnSA-5/6 samples was negligible, whereas a clear increase in aggregates was observed in the HexaPro sample (Fig. 3A–C, right). This trend was also reflected by time-dependent development of PI values at 37 °C (Fig. 3D, right). Curiously, AnSA-5/6 samples appeared less homogenous (higher PI values) and showed higher levels of aggregation when stored at 4 °C than at warmer temperatures. In order to exclude that aggregates were formed due to protein truncation, a separate shelf-life experiment was performed, in which samples of HexaPro, AnSA-5 and AnSA-6 were incubated at 4 °C for 21 days; the temperature associated with highest amount of aggregates for AnSAs. SDS-PAGE analysis showed that none of the proteins contained truncation products (gels provided in Source data).

Next, we wondered if antibodies raised against S protein by natural infection with SARS-CoV-2 were able to bind AnSAs. To this end, we examined the binding of AnSA-5 and AnSA-6 to five human convalescent plasma samples (Supplementary Fig. 8A), including two from vaccine breakthrough infection. The patients likely encountered different variants of SARS-CoV-2 since infection occurred during different points in time (Supplementary Fig. 8A). Two pre-pandemic plasma samples were used as negative controls.

Both AnSA-5 and AnSA-6 reacted to human convalescent plasma samples, albeit with lower relative potency compared to HexaPro (Fig. 4A). The lower sensitivity compared to HexaPro may result from

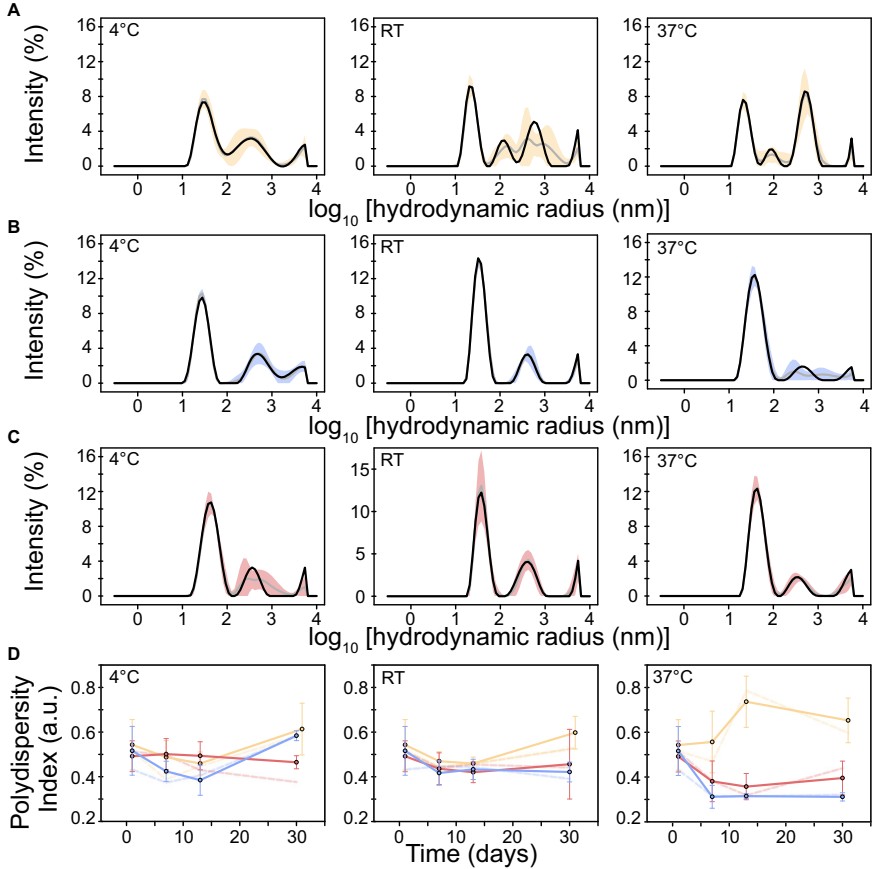

**Fig. 3 | DLS analysis of HexaPro, AnSA-5 and AnSA-6 after storage at different temperatures.** DLS particle size distributions of 1.2–1.5 mg mL⁻¹ **A** HexaPro, **B** AnSA-5 and **C** AnSA-6 samples plotted by intensity. The datasets were measured after one-month storage at the indicated temperature. Average values obtained from five technical repeat measurements are plotted as black lines (calculated by ZS XPLORER software) or grey lines (calculated manually from the individual plots), respectively. Standard deviations from technical variation ($n = 5$) are indicated as coloured area plots (yellow−HexaPro, blue−AnSA-5, red−AnSA-6). **D** Development of polydispersity indices during storage. At the indicated time points, 7 μL of each sample were analysed by DLS and then transferred back into the storage tube. Average PI-values were obtained from five technical repeat measurements and are plotted as solid coloured line and circles (error bars indicate standard deviation, $n = 5$). Dotted lines indicate PI values as derived from an average DLS dataset calculated by ZS XPLORER software.

differences in RBD sequence between AnSAs and the HexaPro construct (Supplementary Fig. 2), preventing antibodies that are specific for linear wt-RBD epitopes to recognize the AnSAs. Moreover, the fact that AnSA-5/6 reside in an all-RBD-down conformation (Fig. 2 and Supplementary Fig. 5) may contribute to loss of RBD recognition of some antibodies in the plasma samples. Nevertheless, these results suggest that antibodies elicited against the S protein, either by natural infection with SARS-CoV-2 or by combination of vaccination and natural infection, are able to recognize AnSAs; likely by binding to conserved regions or to regions in which sequence variation retains the general physiochemical properties of a surface to enable antibodies to cross-react to the modified surface. Both types of antibodies are of interest, as they may be able to bind to S protein of emerging variants of SARS-CoV-2.

The notion that AnSAs adequately represent the structure of conserved epitopes is further supported by the fact that AnSA-5 and -6 bound to CR3022 IgG1 with similar apparent affinity as HexaPro, as studied by surface plasmon resonance (SPR) technology (Fig. 4B). The CR3022 antibody, originally identified for its binding to SARS-CoV[31], is known to engage a non-linear epitope on the RBD that is conserved between the S proteins of SARS-CoV and SARS-CoV-2 and is not involved in binding the ACE2 receptor[32]. Out of 21 residues in the SARS-CoV-2 S protein that were identified to interact with CR3022[32], 18 and 19 residues are conserved in AnSA-5 and AnSA-6, respectively. Despite the fact that the ability of CR3022 to neutralize SARS-CoV-2 S protein in vitro is not fully clear[32–34], its importance in humoral immune response has been identified[32].

We also assessed whether AnSAs interact with the major host cell receptor ACE2. While HexaPro showed a clear interaction with soluble ACE2 in an SPR experiment, no interaction could be observed between AnSA-5/6 and ACE2 (Fig. 4C). A surrogate virus neutralization (sVNT) assay using HexaPro, AnSA-5 and -6 supported this observation (Supplementary Fig. 8B). While a convalescent plasma sample neutralized HexaPro, the neutralization efficiency against AnSAs could technically

not be determined due to the lack of interaction between ACE2 and AnSAs prior to antibody-mediated displacement (no signal in positive control).

The absence of AnSA-ACE2 interaction may be attributed to the change in RBD sequence; 50 and 31 out of 71 residues in the RBD (residues 437–508 in SARS-CoV-2 S protein) differ between AnSA-5/-6 and Wuhan wt S protein sequence, respectively. These sequence differences probably result from the topology of the inferred phylogeny that does not take recombination into account. Specifically, the S protein of Bat SARS-like coronavirus ZC45 shares high overall sequence similarity with SARS-CoV-2 S protein, and therefore localizes to a neighbouring branch of SARS-CoV-2 S protein (Supplementary Fig. 1), whereas its RBD is more closely related to non-ACE2-binding S-proteins that differ from SARS-CoV-2 S protein by several point mutations and deletions. These differences are reflected in the reconstructed AnSA sequences (Supplementary Fig. 2) and likely interfere with ACE2 binding.

### AnSA-5 potently boosted immune responses in vitro

In order to confirm that AnSAs do not only bind to convalescent patient samples but are capable of eliciting a human immune response, we evaluated AnSA-5 immunogenicity using a recently established in vitro tonsillar organoid culture model that has been shown to recapitulate key germinal centre immune responses after immunization with antigens (Fig. 5A)[35]. The tonsil organoid model employed in the present study is an immune organoid model[35], different from conventional organoid models derived from stem cells[36]. HexaPro was used as a reference in the experiments. Although the magnitude of the immune response of different donors largely depended on individual differences such as vaccination/ infection history, age, sex, etc. (Supplementary Table 2), broad humoral immune responses triggered by antigenic stimulation with HexaPro and AnSA-5 could be observed in tonsil organoids derived from three different donors.

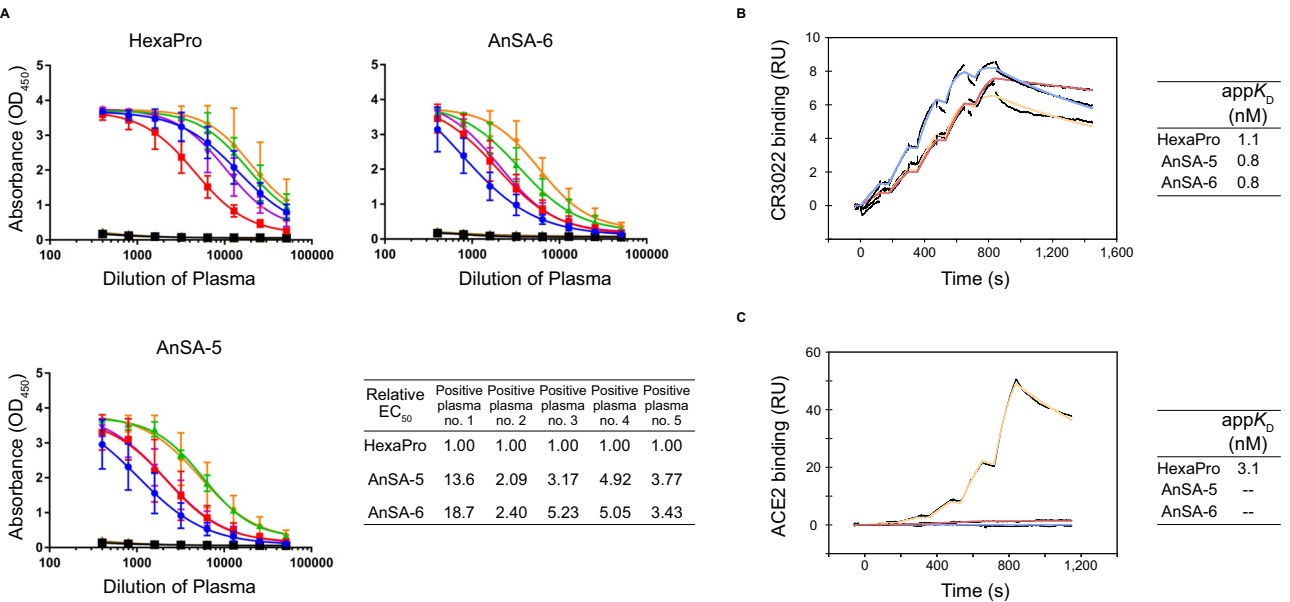

**Fig. 4 | Antigen properties of AnSA-5 and AnSA-6 compared to HexaPro.**
**A** Binding to convalescent human plasma, measured by enzyme-linked immunosorbent assay. COVID-19 convalescent plasma samples ($n = 5$): 1 (blue circles), 2 (red squares), 3 (green triangles), 4 (purple triangles), 5 (orange diamonds). Negative (pre-pandemic) plasma samples ($n = 2$): 1 (black squares), 2 (brown triangles). The results in the graphs are presented as mean and standard deviation of three independent experiments. The EC50 values were determined using a four-parameter non-linear regression and the relative EC50 values were obtained by normalizing

the EC50 value of each AnSA to that of HexaPro. **B** S protein binding to CR3022 human IgG, measured by SPR (single cycle kinetics). Solutions of CR3022 (3.8 nM, 7.5 nM, 15 nM, 30 nM, 60 nM) were flowed over immobilized HexaPro (yellow line), AnSA-5 (blue line) and AnSA-6 (red line). **C** S proteins binding to soluble ACE2 receptor, measured by SPR (single cycle kinetics). Solutions of ACE2 (0.4 nM, 1.1 nM, 3.3 nM, 10 nM, 30 nM) were flowed over immobilized HexaPro (yellow line), AnSA-5 (blue line) and AnSA-6 (red line).

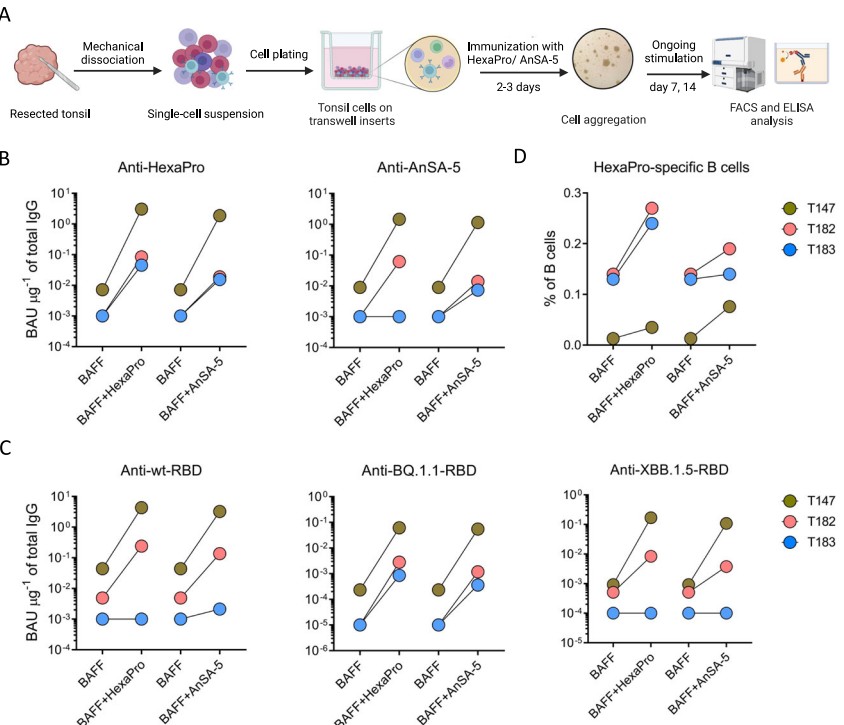

**Fig. 5 | Adaptive immune responses in tonsil organoids stimulated by AnSA-5 and HexaPro. A** Workflow for tonsil tissue disruption, culture preparation, antigen immunization, and functional readouts. Figure created with Biorender.com. **B** Levels of HexaPro- and AnSA-5-binding IgG from day 14 organoid cultures (n = 3, T147, T182 and T183) stimulated with HexaPro or AnSA-5 compared to BAFF control. **C** Levels of wt, BQ.1.1, and XBB.1.5 RBD-binding IgG from day 14 organoid cultures stimulated with HexaPro or AnSA-5 compared to BAFF control. In (**B**) and

(**C**), the antibody titres in the graphs are presented as mean (represented by dots) of measurement of duplicates in one experiment. Data shown as binding antibody unit (BAU) per µg of total IgG detected in tonsil supernatants (BAU µg⁻¹ of total IgG). **D** HexaPro-specific B cell percentages out of CD19⁺ CD20⁺ B cells between BAFF, BAFF and HexaPro, and BAFF and AnSA-5 stimulations from day 14 organoid cultures defined by flow cytometry. Each dot on the graph represents the percentage for a single measurement. In (**B**–**D**), colours represent different donors.

Following seeding of cells in the transwell, the formation of isolated cell clusters was visible within the first three days and depended on the presence of recombinant human B cell-activating factor (BAFF), a B cell survival factor (Supplementary Fig. 9). For two of the three donors (T182 and T183), some of the plasmablast formation over time might be attributed to the addition of BAFF, as indicated by the expansion of plasmablasts from day 7 to 14 in the BAFF control group for these donors (Supplementary Fig. 10A, middle and right). However, the slightly stronger decrease in the pre-germinal centre (pre-GC) pool of B-cells from day 7 to 14 in the antigen-stimulated groups compared to the respective BAFF control groups for these two donors (Supplementary Fig. 10A, middle and right) indicates an antigen-induced immune response. This observation is further supported by the greater increase in the plasmablast pool in the antigen-stimulated groups compared to the respective BAFF control groups (Supplementary Fig. 10A, middle and right) as well as the increase of IgG titres in antigen-stimulated groups compared to the respective BAFF control groups (Fig. 5B, C).

Next, purified HexaPro was used as a proxy in order to assess the activity of antigen-induced immune responses against SARS-CoV-2 S protein. HexaPro-specific B cells (Fig. 5D) and IgG antibodies (Fig. 5B, left) were detected in all donors 14 days after the addition of either of the two antigens, HexaPro and AnSA-5. Specifically, all donors exhibited more HexaPro-specific B cells in the two antigen-stimulated groups compared to the BAFF control group (Fig. 5D), which are mostly of the plasmablast phenotype (Supplementary Fig. 10B); in line with higher HexaPro-binding antibody titres induced by HexaPro and AnSA-5 compared to the BAFF control group (Fig. 5B, left). Noteworthily, although the HexaPro-specific B cell populations were phenotypically diverse across donors, donor T147 showed the highest IgG titres in response to HexaPro stimulation compared to the other two

donors (Fig. 5B, left); probably reflecting the sole generation of HexaPro-specific plasmablasts (Supplementary Fig. 10B). Finally, T cell population dynamics were relatively unchanged (Supplementary Fig. 10C) between donors and across conditions.

Longitudinal ELISA results (Supplementary Fig. 11) revealed that tonsil organoid immunization with AnSA-5 induced both anti-HexaPro and anti-AnSA-5 antibodies in a time dependent manner (blue lines in Supplementary Fig. 11, left and middle). This may be explained by the hypothesis that AnSA-5 immunization potently recalled the established immune memory from previous infection or vaccination among the donors by recognizing epitopes that are shared among HexaPro, AnSA-5 and the SARS-CoV-2 S protein that the donors were pre-immunized with. The titres of anti-HexaPro antibodies elicited by HexaPro and AnSA-5 were of the same magnitude (Fig. 5B, left), highlighting that the two antigens have a comparable capacity of inducing S protein binding antibodies. Moreover, in donor T183, anti-AnSA-5 antibody was solely detected in the AnSA-5 stimulated group but not the HexaPro-stimulated group, showing that a de novo AnSA-5 specific immune response was triggered in an antigen-specific way for this donor (Fig. 5B, right).

In order to assess the utility of AnSAs in inducing cross-reactive antibodies, we further compared antibody levels against the Wuhan wt-RBD as well as the RBD of two circulating variant of concerns (VOCs) Omicron BQ.1.1 and XBB.1.5, as triggered by HexaPro and AnSA-5 immunization. Notably, AnSA-5 induced not only broad-spectrum but also high-titre antibody responses against VOCs on par with HexaPro, at least for two donors, indicating its potential as a promising boost vaccine candidate to fight against future emerging VOCs (Fig. 5C, middle and right).

We subsequently measured the neutralization activity against the wt SARS-CoV-2 pseudovirus in the culture supernatant of immunized

tonsil organoids. We detected neutralization activity for donor T147 immunized with AnSA-5 (50% neutralization titre (NT$_{50}$): 14.05) and HexaPro (NT$_{50}$: 232.5) as well as donor T182 immunized with AnSA-5 (NT$_{50}$: 7.29) (Supplementary Fig. 12). For T183, the low titre of anti-RBD antibodies, as measured in ELISA (Fig. 5C, left panel), were not sufficient to detect neutralization in the pseudovirus assay. These results thus suggest that the AnSA-5 protein can induce neutralizing antibodies against the wt S protein.

## Using AnSAs as scaffold antigens

Despite the fact that the RBD sequence of AnSA-5 and -6 differs from the wt-RBD sequence (28 and 23% sequence difference to the wt-RBD sequence, respectively), the overall RBD tertiary structure is similar (Cα RMSD of 1.00 and 1.10 Å across 157 and 159 pruned atom pairs for AnSA-5 and -6 to wt-RBD (PDB-ID 6VXX)[26], respectively). We therefore exchanged the AnSA-RBD with the equivalent residues of wt-RBD sequence and wondered whether the exchange would restore ACE2 binding in the AnSAs.

AnSA-5 and -6 that harbour the wt-RBD could be purified with similar purity but somewhat lower yield than AnSA-5/6 and HexaPro (Supplementary Fig. 13A). The DLS profile of AnSA-5 harbouring the wt-RBD showed a similar amount of aggregates compared to AnSA-5, whereas the aggregate peak increased for AnSA-6 harbouring the wt-RBD (Supplementary Fig. 13B, C, PI values 0.60 and 0.83, respectively). The DLS peak present in the mid-range size (ca. 150 nm) that was observed in HexaPro (Supplementary Fig. 3D) is clearly present in the AnSA-6 wt-RBD DLS trace as well. In summary, these results indicate AnSA-5 as a more suitable scaffold for RBD exchange. Exchanging the RBD in AnSA-3 expectedly resulted in negligible yields (Supplementary Fig. 13A), as the starting titres for AnSA-3 were low (Supplementary Fig. 3A), which is why we did not proceed with this variant. Despite the fact that the expression yields of wt-RBD scaffolded antigens (AnSA-5/6) are lower than the AnSAs themselves, they are superior to AnSA-3 and the reported yield of the S2P variant[14].

The DSF data of AnSA-5/6 harbouring the wt-RBD are largely consistent with the DLS results. AnSA-5 harbouring the wt-RBD seems somewhat stable, with $T_m$ values of 49 and 61 °C (blue line in Supplementary Fig. 13D). The $T_m$ value associated with the first peak (49 °C, RBD domain unfolding) is comparable to both HexaPro (47 °C) and AnSA-5 (50 °C), whereas the $T_m$ value associated with the second peak (61 °C, trimer unfolding) is comparable to HexaPro (63 °C) but decreased compared to AnSA-5 (67 °C). Likewise, AnSA-6 harbouring the wt-RBD was observed to lose some thermostability. This variant exhibits an unclear melting profile with a major melting event between 45 and 65 °C (red line in Supplementary Fig. 13D), which may indicate the disintegration of the protein structure, potentially due to shedding of the S1 subunit.

Most mutations in AnSA-5 and -6 compared to the SARS-CoV-2 S protein are located in the NTD and to a lesser extent in the RBD (42% and 28% sequence difference to the wt domains, respectively for AnSA-5, Supplementary Fig. 2). This reflects the fact that the sequences that were included in the phylogenetic tree show largest sequence differences within the NTD domain. In the AnSA trimer structures the three RBDs pack closely against each other (Fig. 2A, top and Supplementary Fig. 5) and against the NTD and loop motifs (Fig. 2A, bottom and Supplementary Fig. 5), as facilitated by extensive hydrogen bond networks across the domain interfaces. By re-inserting the wt-RBD into the scaffold antigens, such hydrogen bonds are likely broken, which would explain a loss of stability. Nevertheless, the stability of AnSA-5 harbouring the wt-RBD is still comparable to HexaPro and increased compared to S2P.

Exchanging AnSA-5/6-RBDs for the wt-RBD restored receptor binding to the ancestral S proteins (apparent $K_D$-values of 2.4 nM and 2.6 nM, respectively, Fig. 6A) with similar apparent affinity as HexaPro (3.1 nM, Fig. 4C). These results indicate that the ancestral scaffolds are able to host the wt-RBD domain correctly folded for presentation to the ACE2 receptor. Moreover, these chimeric S proteins bound to several recombinant monoclonal antibodies raised against RBD/S1

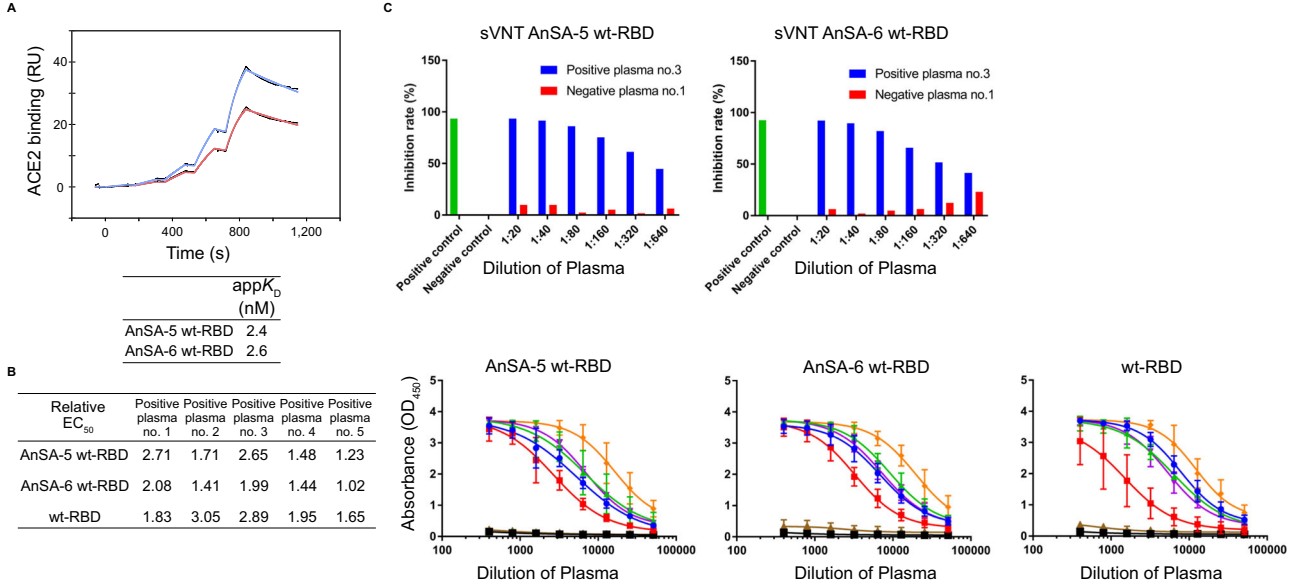

**Fig. 6 | Antigen properties of AnSA-5 and AnSA-6 harbouring the wt-RBD sequence. A** Binding of AnSAs harbouring the wt-RBD to soluble ACE2 receptor, measured by SPR (single cycle kinetics). Solutions of ACE2 (0.4 nM, 1.1 nM, 3.3 nM, 10 nM, 30 nM) were flowed over immobilized AnSA-5 wt-RBD (blue line) and AnSA-6 wt-RBD (red line). **B** Binding of AnSA-5/6 harbouring wt-RBD as well as the isolated wt-RBD domain to convalescent human plasma, measured by enzyme-linked immunosorbent assay. COVID-19 convalescent plasma samples (*n* = 5): 1 (blue circles), 2 (red squares), 3 (green triangles), 4 (purple triangles), 5 (orange diamonds).

Negative (pre-pandemic) plasma samples (*n* = 2): 1 (black squares), 2 (brown triangles). The results in the graphs are presented as mean and standard deviation of three independent experiments. The EC50 values were determined using a four-parameter non-linear regression and the relative EC50 values were obtained by normalizing the EC50 value of each protein to that of HexaPro (Fig. 4A). **C** Surrogate virus neutralization assay of AnSA-5/6 wt-RBD. Each column on the graph represents the percentage of inhibition for a single measurement for one positive (no. 3, blue column) and one negative (no. 1, red column) plasma sample.

domains of the Wuhan wt S protein (Supplementary Fig. 14) and to antibodies in convalescent plasma samples (Fig. 6B) with EC50 values that are lower than both the respective AnSAs (Fig. 4A) as well as slightly lower than isolated wt-RBD (Fig. 6B, right).

Some of these antibodies are likely neutralizing, as indicated by the fact that a plasma sample was able to inhibit the interaction of AnSA-5/6 harbouring the wt-RBD with ACE2 (Fig. 6C). These results highlight the feasibility of exchanging AnSA-5/6-RBDs with wt-RBD, thus creating chimeric S proteins that benefit from robust ancestral folds. While this exchange comes at the cost of yield and stability compared to the parental AnSA proteins, they are still favourable compared to the wt or S2P variant.

## Discussion

A major bottleneck limiting the full potential of biopharmaceuticals is low protein stability, reducing both shelf-life and potency, as in vivo residence time is linked to immunogenic potential[37]. Engineering to improve vaccines and other biopharmaceuticals is commonly executed by cycles of structure-guided single amino acid substitutions. For COVID-19, several vaccines currently approved for use worldwide are based on rational mutations in S protein to introduce two prolines[19], without which the stability of the protein is prohibitively low. These two proline residue substitutions, originally identified from structure-based protein engineering efforts on S protein from MERS-CoV and SARS-CoV, enabled stabilization of SARS-CoV-2 S protein to a level that allow for structural elucidation of the full ectodomain. Only from this stage could additional engineering approaches be implemented to further improve expression yields and stability of the antigen. In the context of a pandemic, it is not always given that previous single-residue stabilization techniques are known or readily applicable to a novel virus, and structural information to guide protein engineering campaigns can be lacking.

Herein we reconstructed ancestral betacoronavirus spike protein scaffold antigens independent of structural input. ASR has been used in biopharmaceutical research[38,39], still its use in antigen development has not been fully explored[40]. The ancestral S proteins presented herein showed enhanced stability, solubility and expression yields compared to SARS-CoV-2 S protein and bound antibodies in plasma of convalescent COVID-19 patients. Furthermore, an ancestral S protein stimulated the production of SARS-CoV-2 S protein binding as well as cross-reactive antibodies in tonsil organoids. They do not contain the two proline substitutions and differ from SARS-CoV-2 mainly by mutations on the protein surface. Moreover, AnSAs did not show an elevated tendency to aggregate even after storage at 37 °C for 4 weeks. Replacing the ancestral RBDs in AnSA-5 and AnSA-6 to the wt-RBD sequence recovered the ability of the ancestral proteins to bind to the ACE2 receptor and increased serum activity. This finding demonstrates that AnSA-5 and AnSA-6 are scaffold antigens for which domains of novel coronaviruses may be grafted into the stable ancestral backbone. The lack of general methods to rapidly design stable protein-based antigens amenable for distribution in low-income nations has been identified as a major hurdle obstructing equitable access to vaccines in the ongoing and future pandemics[41]. Given that only four gene constructs were tested in the lab, the AnSA-based approach described herein represents a straightforward and low throughput way to generate potent antigens from available sequence space, without the need to test or screen variant libraries.

## Methods

### Ethics statement

Our research complies with all relevant ethical regulations. The sampling of blood was performed under the approval of the Institutional Review Board of Policlinico San Matteo (protocol number P_20200029440) and the ethics committee in Institutional review board in Stockholm (Dnr 2022-00676-01). The collection of tonsils was performed under the approval of Institutional review board in Stockholm (Dnr 2023-02803-01). The participants provided written informed consent before participation in the study. Information regarding age, sex, and history of COVID-19 vaccination and infections was collected from each participant through a questionnaire. The parent or legally authorized representative provided informed consent on behalf of participants under the age of 18 and the requested information for the study. No compensation was provided to participants.

### Ancestral sequence reconstruction

The target amino acid sequence of wt SARS-CoV-2 S protein (Uniprot P0DTC2) was used to find homologous sequences using a BLAST-search. Since many betacoronavirus S protein sequences available in databases originate from clinical isolates of SARS-CoV and MERS-CoV with marginal differences to the wt sequences, only a limited number of these sequences that originate from distinct species were included for tree building, in order not to bias the alignment. The top 250 sequences retrieved from BLAST with highest sequence similarity to the target sequence were subsequently aligned in MEGA-X[42] v10.1.6 using the MUSCLE algorithm[43]. Duplicate sequences and single point variants of the same sequences were excluded from the alignment. Subsequently, a maximum likelihood phylogenetic tree was constructed in IQ-Tree[44] v1.6.12 using a WAG + F + R8 substitution model. Branch supports were obtained using ultrafast bootstrap[45] with 1000 replicates. From the tree and alignment, the ancestral sequences were reconstructed using the maximum likelihood ancestral inference option in MEGA-X with a gamma distribution to account for variable mutation rates. Four amino acid sequences corresponding to ancestral nodes upstream of SARS-CoV-2 S protein (nodes 3, 5, 6 and 9, referred to as AnSA-3, AnSA-5, AnSA-6 and AnSA-9, respectively) were selected for expression and analysis.

### Gene constructs

The obtained ancestral sequences were aligned with the wt SARS-CoV-2 S protein. The parts of the ancestral sequences aligning with the ectodomain of wt SARS-CoV-2 S protein (residues 14–1208) were reverse translated to nucleotide sequences using a codon table for optimized expression in human embryonic kidney cells. The nucleotide sequence of the signal peptide of SARS-CoV-2 S protein (amino acid residues 1–13) was added upstream of the gene and the nucleotide sequence of a GS-linker and T4-Foldon trimerization domain were added downstream of the gene (nucleotide sequences obtained from pαH vector encoding the HexaPro protein[14]). The final gene constructs were obtained in a pMx-series vector from Invitrogen GeneArt services (ThermoFisher Scientific, USA) and cloned into a pαH vector encoding the HexaPro protein[14] using *BamHI* and *SpeI* restriction sites to replace the HexaPro sequence with the respective ancestral gene constructs. The resulting gene constructs consist of the wt signal peptide, the ancestral S protein sequence that corresponds to the wt ectodomain, a GS-linker, a T4-Foldon-domain, a GTS-linker, the HRV 3 C protease recognition peptide, a G-linker, a His$_8$-tag and a Twin-Strep-tag® and were confirmed by Sanger sequencing (Eurofins Genomics, Germany).

The wt-RBD domain (corresponding to residues 319–541) was produced from a pCAGGS vector carrying the gene with a C-terminal His$_6$-tag (BEI resources, USA).

For the generation of AnSA-5 and -6 harbouring SARS-CoV-2 wt-RBD, the wt-RBD gene was amplified from a pCAGGS vector (BEI resources) and cloned into the pαH vector carrying the ancestral gene constructs using Golden Gate assembly, replacing the corresponding ancestral RBD sequences. The final constructs were confirmed by Sanger sequencing (Eurofins Genomics, Germany).

Soluble ACE2 protein was produced from a pFUSE$_{2SS}$-CLIg-hK plasmid carrying the gene with a C-terminal HSA-cIFN- tag[46].

## Protein expression

The ancestral S proteins were expressed using the Expi293 expression system (ThermoFisher Scientific, USA) according to the manufacturer's instructions. Expi293F cells were cultivated in the manufacturer's expression medium at 37 °C at 115 rpm with 8% $CO_2$ at 80% humidity. The cells were grown in small scale (50 mL) in 250 mL non-baffled flasks (Nalgene, USA) or large scale (1 L) in 2.8 L non-baffled flasks (Nalgene, USA). In order to check cell viability, cells were stained with Trypan Blue and counted using a CELENA® S Digital Imaging System. Cells were split to $0.8 \times 10^6$ cells mL$^{-1}$ the day before transfection. For transient transfection polyethylenimine (PEI) and DNA (1 µg plasmid DNA per $10^6$ cells) were combined in a 1.5:1 (w/w) ratio. The PEI-DNA mixture was incubated at room temperature for 20 min before being added dropwise to a culture at a cell density between $1.2–1.8 \times 10^6$ cells mL$^{-1}$. Transfected cells were left in the incubator under aforementioned conditions for 3 days prior to cell harvesting and protein purification. For experiments assessing protein production yields at different transfection durations, cells were transfected for 3–5 days.

## Protein purification

For purification of HexaPro, AnSAs and the isolated wt-RBD, cell cultures were harvested by centrifugation at $4000 \times g$ at 4 °C and the supernatant was filtered through Rapid-Flow bottle top filters (0.2 µm pore size, ThermoFisher Scientific, USA). Supernatant from large scale cultures was concentrated to a volume of ca. 100 mL using the Vivaflow 200 Laboratory Cross Flow Cassette (Sartorius, Germany). Supernatant from small scale cultures was kept at the original volume. The supernatant was then incubated with Ni-NTA resin (Qiagen, Germany) that had been pre-equilibrated with 10 column volumes (CV) of wash buffer at a 1:100 (v/v) ratio in end-over-end rotation at 4 °C overnight. The wash buffer used was 20 mM 4-(2-hydroxyethyl)-1-piperazineethanesulfonic acid (HEPES), pH 7.5, 200 mM NaCl. The next day, the solution was passed through an EconoPac chromatography column (Bio-Rad, USA) at 4 °C by gravity flow. The resin was washed with 5 CV wash buffer at 4 °C. Subsequently, the protein was eluted using $3–5 \times 1$ CV of elution buffer (20 mM HEPES pH 7.5, 200 mM NaCl, 250 mM imidazole). The purity of purification fractions was verified by SDS-PAGE using 4–15% Mini-PROTEAN™ TGX Stain-Free™ Protein Gels (Bio-Rad, USA). The fractions were concentrated to a volume of ca. 400 µL in an Amicon Ultra centrifugal spin filter (100 kDa molecular weight cut-off, Merck Group, Germany) that had previously been equilibrated with 15 mL wash buffer. For large scale purifications the protein was further purified by gel filtration using a Superdex 200 Increase 10/300 GL column (Cytiva, USA, formerly GE Health Care, Sweden) in an Agilent 1220 liquid chromatography system using a flow rate of 0.3–0.4 mL min$^{-1}$ in 100 % wash buffer. Elution fractions of 400 µL were collected and their purity was checked by SDS-PAGE. Fractions corresponding to trimeric spike protein (as assessed by the purification UV chromatogram) were collected and concentrated as described above. For small scale purifications, Ni-NTA elution fractions were desalted into wash buffer by gravity flow using PD-10 columns (Cytiva, USA) and were concentrated as described above.

For purification of soluble ACE2 receptor[46], 400 mL of filtered cell culture supernatants were buffer-exchanged into phosphate-buffered saline (PBS) at pH 7.4 and concentrated to a volume of ca. 5 mL. The solution was passed over 2.5 mL of CaptureSelect™ Human Albumin Affinity Matrix (ThermoFisher Scientific, USA) that had been pre-equilibrated with 10x CV of PBS, pH 7.4 on an EconoPac chromatography column by gravity flow. The matrix was washed with $2 \times 5$ CV of PBS, pH 7.4 and protein was eluted with $5–10 \times$ CV of elution buffer (20 mM tris(hydroxymethyl)aminomethane (Tris), pH 7.4, 2.0 M $MgCl_2$). The protein was subsequently purified by gel filtration chromatography using 20 mM HEPES, pH 7.5, 200 mM NaCl and concentrated as described above.

All protein concentrations were determined spectrophotometrically using calculated molar extinction coefficients on an Implen NanoPhotometer NP80 (Germany) and the purified proteins were snap-frozen in liquid nitrogen and stored at −80 °C.

## Dynamic light scattering

Dynamic light scattering plots were recorded on a Zetasizer Ultra (Malvern Panalytical, UK) using 7 µL of protein solution in a low volume disposable capillary that was mounted inside the ZSU1002 capillary cell. Each sample was run with five consecutive technical repeat measurements. The material was set to protein (refractive index 1.59, Abs 0.01) and the dispersant was set to water (refractive index 1.33, viscosity 0.8872 mPa*s. at 25 °C). Polydispersity indices were calculated from the intensity distribution plots using the instrument software (ZS Xplorer v3.0.0.53). For assessing DLS traces during one-month storage, the sample was transferred back into the storage tube that was sealed with parafilm and kept at 4 °C, room temperature or 37 °C until the next time point was measured.

## Cryo-EM

S protein trimer AnSA-5 (1.5 mg mL$^{-1}$) and AnSA-6 (1.5 mg mL$^{-1}$) were diluted to a final concentration of 0.3 mg mL$^{-1}$ in a buffer (20 mM HEPES pH 7.5, 100 mM NaCl). Prior to cryo-EM grid preparation, grids were glow-discharged with 20 mA for 60 s in a GlowQube system (Quorum, UK). Grids used were UltrAuFoil Gold 300 mesh (R 0.6/1 geometry; Quantifoil Micro Tools GmbH, Germany; VHH VE). A volume of 3 µL of sample was applied to the grid and then vitrified in a Vitrobot Mk IV (Thermo Fisher Scientific) at 4 °C and 100% humidity (blot 3 s, blot force 0.595 filter paper (Ted Pella Inc., USA)). Cryo-EM data collection was performed with EPU (Thermo Fisher Scientific) using a Krios G3i transmission-electron microscope (Thermo Fisher Scientific) operated at 300 kV. Movies were acquired at a nominal magnification of 105k with a pixel size of 0.833 Å in nanoprobe EFTEM SA mode with a slit width of 20 eV using a K3 Bioquantum for 2.8 s, during which 45 movie frames were collected with a fluency of 1.11 e$^-$ Å$^{-2}$ per frame (Supplementary Table 1). CryoSPARC v3.3.1 was used for all data processing[47].

A total of 5608 and 12957 micrographs for AnSA-5 and AnSA-6 respectively were selected based on an estimated resolution cut-off of 6 Å. An initial small set of micrographs was used for blob picking of particles that were used to generate a set of 2D class references that were used for template-based picking of 2,555,520 and 2,035,826 particles for AnSA-5 and AnSA-6, respectively. Particles were extracted with a box size of 512 pixels and Fourier cropped to 128 pixels and used for 2D classification. 2D classes with distinct features were selected for future processing, including 1,085,028 and 528,939 particles for AnSA-5 and AnSA-6, respectively. Initial 3D reconstruction was performed ab initio with cryoSPARC Live with one class using a selection of 50,000 particles. Heterogeneous refinement was performed with three classes for AnSA-5 and two classes for AnSA-6 (using corresponding number of copies of the initial ab initio 3D reconstruction as templates), leading to two significantly superior classes for AnSA-5 and one significantly superior 3D class for AnSA-6, that were used for the final 3D reconstruction of 959,111 and 253,429 particles for AnSA-5 and AnSA-6, respectively (Supplementary Fig. 4). Homogeneous refinement produced a 3D reconstruction at an overall resolution 2.59 and 2.77 Å for AnSA-5 and AnSA-6, respectively. Refinements were performed with C1 symmetry and Fourier cropping to 1.11 Å per pixel was used for the final map. Global CTF refinement with global aberration correction (beamtilt, trefoil, tetrafoil and spherical aberration) were performed within the homogenous refinement step.

## Cryo-EM model building and refinement

The models for AnSA-5 and AnSA-6 were generated in ChimeraX v1.3[48] using ISOLDE v1.1[49] with PDB-ID 6ZOZ[16] and PDB-ID 7BNN[50] as starting

models. Coot v0.8.9.1[51] was used to build in missing residues and loop regions. The final models were real-space refined in Phenix v1.3[52] (Supplementary Table 1). Figures were generated using Chimera X v1.3.

## Differential scanning fluorimetry (DSF)

Melting curves for HexaPro, AnSA-5 and AnSA-6 were recorded on a Bio-Rad CFX system (C1000 Touch™ Thermal cycler). 1 µL of 25x SYPRO™ Orange Protein gel stain solution (Thermo Fisher Scientific, USA) was mixed with 8–10 µg of protein and the final volume was adjusted to 25 µL using wash buffer in 96 well Multiplates™ (Bio-Rad, USA). The plates were sealed using optical assay sealing film (Bio-Rad, USA), vortexed and centrifuged (600 × g, 1 min) to remove bubbles. Subsequently, thermal unfolding was monitored on the FRET channel between 20 °C and 95 °C at a temperature gradient of 1 °C min⁻¹. Melting curves were analysed using the CFX Manager software (Bio-Rad, USA) and the derivative of the fluorescence signal was plotted as function of temperature.

## NanoDSF

Melting curves were recorded using a Prometheus NT.48 nanoDSF instrument (NanoTemper Technologies, Germany). Proteins were diluted to 2 mg mL⁻¹ in wash buffer ("reference buffer") or in wash buffer containing either 2 M urea or 2 M guanidine hydrochloride. Ca. 10 µL of protein solution were soaked into glass capillaries and thermal unfolding was monitored as change in the ratio of protein intrinsic fluorescence at 350 and 330 nm (F350/F330) between 20 and 90 °C at a temperature gradient of 1 °C min⁻¹. Melting curves were analysed using the nanoTemper software (PR.ThermControl v2.3.1) and the derivative of the fluorescence ratio was plotted as function of temperature. The values of the first derivative were normalized between −1 and +1 for each protein separately (taking into account all tested buffer conditions). Backscattering data were recorded during the same runs to assess onset of aggregation of larger particles and both scattering data and the first derivative of scattering data were plotted as function of temperature.

## Plasma binding assays

The participants ($n = 7$), aged 37–83, included 4 males and 3 females and were randomly selected among cohorts in Sweden and Italy. The recruitment criteria for collection of blood from convalescent patients included having a documented history of COVID-19 infection (severity and days after infection) and vaccination (type of vaccine, number of doses, the interval between infection and vaccine doses, days after the latest dose, and breakthrough infection), and who were willing and able to provide written informed consent. Five SARS-CoV-2 S protein- and RBD-specific antibody positive plasma samples were obtained from convalescent patients including two with breakthrough infection following two doses of Pfizer vaccine. Negative sera controls were acquired from two pre-pandemic health donors (Supplementary Fig. 8A)[53,54]. The study was performed under the approval of the Institutional Review Board of Policlinico San Matteo (protocol number P_20200029440) and the ethics committee in Stockholm (Dnr 2022-00676-01). To examine the antigenicity of S protein variants, AnSA-5, AnSA-6, AnSA-5 wt-RBD, AnSA-6 wt-RBD, HexaPro and isolated wt-RBD were coated (1.7 µg mL⁻¹ in PBS) on high-binding Corning Half area plates (Corning #3690, USA) overnight at 4 °C. After blocking with 2% bovine serum albumin (BSA) for 1 h, serial dilutions of plasma in 0.1% BSA in PBS were added and plates were subsequently incubated for 1.5 h at room temperature. Plates were then washed and incubated for 1 h at room temperature with horseradish peroxidase-conjugated goat anti-human IgG (Invitrogen #A18805, USA) (diluted 1:15000 in 0.1% BSA-PBS). Bound antibodies were detected using tetramethylbenzidine substrate (Sigma #T0440). The colour reaction was stopped with 0.5 M $H_2SO_4$ after 8 min incubation and the absorbance was measured at 450 nm in an ELISA plate reader. All samples were run in the same experiment and the results expressed as $OD_{450}$. The EC50 values were determined using a four-parameter non-linear regression (GraphPad Prism 7.04 software) and the relative EC50 values were obtained by normalizing the EC50 value of each ancestral S protein to that of HexaPro.

## SPR measurements (CR3022 binding, ACE2 binding)

SPR measurements using a single cycle kinetic (SCK) approach were performed on a Biacore 8K instrument (Cytiva, Sweden) at 25 °C in running buffer HBS-EP (10 mM HEPES pH 7.4, 150 mM NaCl, 3 mM EDTA, 0.05% surfactant P20). Ancestral S proteins AnSA-5 and AnSA-6, as well as their wt-RBD counterparts and HexaPro, were each captured onto separate surfaces of a streptavidin series S sensor chip (Cytiva, BR100531) according to manufacturer's recommendations. A three-fold dilution series of ACE2 comprised of five concentrations ranging between 30–0.4 nM were prepared in running buffer. A two-fold dilution series of CR3022 IgG1 (srbd-mab1, InvivoGen, USA) comprised of five concentrations ranging between 60–3.8 nM was also prepared in running buffer. Each dilution series was sequentially injected and allowed to bind to the captured S proteins. Following a dissociation phase, surface regeneration was accomplished by injecting regeneration buffer (1 M NaCl, 50 mM NaOH). By subtracting the response curve of a reference surface (streptavidin coated) and to a blank run (running buffer injected as analyte) response curve sensorgrams were obtained. Data was analysed using the Biacore Insight Evaluation software (Cytiva).

## Surrogate virus neutralization assay

The presence of plasma neutralizing antibodies that prevent the S proteins (AnSA-5, AnSA-6, AnSA-5/6-wt RBD and HexaPro) from binding to ACE2 was evaluated by a surrogate SARS-CoV-2 virus neutralization test (sVNT) kit (ProteoGenix, France) according to the manufacturer's instructions with slight modifications. Instead of using the plate pre-coated with SARS-CoV-2 RBD, we coated plate with ancestral S proteins at 1.7 µg mL⁻¹ in PBS overnight at 4 °C. The positive and negative plasma samples were tested at dilutions from 1/20 to 1/640.

## Organoid model

Whole tonsils were collected from three male individuals (age 4–22 years) undergoing surgery for obstructive sleep apnoea. Matched blood samples were collected to confirm that the donors have been infected or vaccinated by measuring the level of anti-HexaPro antibodies. A written informed consent and self-reported information on COVID-19 infection and vaccination history were obtained from the donors (Supplementary Table 2). The parent or legally authorized representative provided informed consent on behalf of participants under the age of 18 and the requested information for the study. The study was approved by the ethics committee of the institutional review board of Stockholm.

Tonsil organoids were then prepared following the sample preparation method as described by Wagar et al.[35]. Specifically, tonsil tissues were dissected and cell suspensions were prepared, enumerated, and resuspended to $2 \times 10^7$ cells mL⁻¹ [35]. Fresh cells were plated, 100 µL per well, into permeable (0.4-µm pore size) membranes (48-well size PTFE or polycarbonate membranes in standard 24-well plates) with the lower chamber consisting of complete medium (1 mL) supplemented with 0.5 µg mL⁻¹ of recombinant human B cell-activating factor (BAFF; BioLegend #559608) as a control or 0.5 µg mL⁻¹ BAFF and either AnSA-5 or HexaPro at a final concentration of 0.15 µg mL⁻¹. Cultures were incubated at 37 °C, 5% $CO_2$ with humidity.

The supernatant (500 µL) was collected from the lower chamber on days 3, 7, 10, and 14 for assessment of S protein- and RBD-binding antibodies and total IgG by ELISA and replaced with 500 µL fresh

medium supplemented with BAFF. The cells were collected on days 0, 7, and 14 and analysed by flow cytometry.

## Specific antibody and total IgG levels in organoid supernatants

BQ.1.1 (#40592-V08H143) and XBB.1.5 RBD (#40592-V08H146) were purchased from Sino Biological. To assess the S protein- and RBD-binding IgG antibody levels in the culture supernatants, high-binding Corning half-area plates (Corning #3690) were coated with purified HexaPro and AnSA-5, or purified RBD domains of Wuhan wt, BQ.1.1, and XBB.1.5 (1 µg mL⁻¹). Serial dilutions of culture supernatant in 0.1% BSA in PBS were prepared and the ELISA was performed as described above. Specific IgG levels were expressed as binding antibody unit (BAU) mL⁻¹ by calibrating the in-house standards using the WHO International Standard for anti-SARS-CoV-2 immunoglobulin (NIBSC, 20/136). S protein- and RBD-binding IgG antibody levels were further normalized to the total level of IgG (BAU µg⁻¹ of total IgG).

To assess total IgG, high-binding Corning half-area plates (Corning #3690) were coated overnight at 4 °C with polyclonal goat anti-human IgG (Southern Biotech #2040-01) (2 µg mL⁻¹) in PBS. Serial dilutions of culture supernatants were added and the wells were subsequently incubated with HRP-conjugated polyclonal goat anti-human IgG (Invitrogen #A18805) diluted 1:15000. The bound antibodies were detected as described above. Serial dilutions of standard human IgG serum (Karolinska University Hospital) were used for the generation of standard curves and measurement of concentrations (µg mL⁻¹).

## Pseudovirus neutralization assay

The gene encoding the wild-type S protein lacking the C-terminal 19 codons ($S_{\Delta 19}$) was synthesized by GenScript using human codon optimization. To generate (HIV-1/NanoLuc2AEGFP)-SARS-CoV-2 particles, three plasmids were used, with NanoLuc luciferase and EGFP reporter genes (pCCNanoLuc2AEGFP), HIV-1 structural/regulatory proteins (pHIV$_{NL}$GagPol) and SARS-CoV-2 $S_{\Delta 19}$ carried by separate plasmids, as previously described[55,56]. 293FT cells (Invitrogen #R70007) were transfected with 7 µg of pHIV$_{NL}$GagPol, 7 µg of pCCNanoLuc2AEGFP, and 2.5 µg of pSARS-CoV-2-$S_{\Delta 19}$ carrying the $S_{\Delta 19}$ gene from the G614 strain of SARS-CoV-2 (at a molar plasmid ratio of 1:1:0.45) using 66 µl of 1 mg mL⁻¹ PEI.

Twofold serial dilutions of tonsil organoid culture supernatants (1:5–1:640) were incubated with pseudovirus carrying the wt S protein from SARS-CoV-2 for 1 h at 37 °C. The mixture was subsequently incubated with 293T-hACE2 cells (a gift from Paul D. Bieniasz, The Rockefeller University) for 48 h, after which the cells were washed with PBS and lysed with Luciferase Cell Culture Lysis reagent (Promega). NanoLuc luciferase activity in the lysates was measured using the Nano-Glo Luciferase Assay System (Promega) with a Tecan Infinite microplate reader. The relative luminescence units were normalized to those derived from cells infected with pseudotyped virus in the absence of tonsil organoid culture supernatant. The neutralization titre 50 (NT$_{50}$) was expressed as the maximal dilution of the culture supernatant where the reduction of the signal is ≥50%. The experiment was performed in duplicates and the mean neutralization (%) values are presented in Supplementary Fig. 12. The NT$_{50}$ values were determined using four-parameter nonlinear regression (the least squares regression method without weighting) (GraphPad Prism 7.04 software). As a positive control, fivefold dilutions of the monoclonal IgG anti-RBD antibody SA55 (produced by Genscript) ranging from 1 µg mL⁻¹ to 0.01 ng mL⁻¹ were used[57].

## Flow cytometry

Cells from the organoids were collected on days 7 and 14 from the upper part of the permeable membrane, washed with PBS and filtered through a 70-µm cell strainer. In order to detect HexaPro-specific cells, the cells were stained in three consecutive steps: first cells were incubated for 45 min with purified HexaPro at 4 °C, followed by a 30 min

room temperature staining with live/dead Aqua (ThermoScientific, #L34957) and finally Fc-blocked (BD Biosciences, #564219, 1/100) for 10 min and stained with the following anti-human antibodies, all from BD Biosciences unless otherwise stated: FITC CD3 (2/100), PE and APC anti-His Tag antibody (Biolegend; 5/100), PE-Cy7 CD19 (1/50), PE-Cy5 CD20 (1/50), BV786 CD38 (1/50), BUV737 CD27 (1/100). HexaPro was detected via its His$_8$-tag and non-cell-bound HexaPro was removed using extended washing steps.

For the characterization of B cell and T cell subsets, cells were first Fc-blocked at 1/100 (BD Biosciences, #564219), followed by a 30 min room temperature staining with live/dead Aqua as well as the following anti-human antibodies, all from BD Biosciences unless otherwise stated: APC CD8α (1/100), FITC CD4 (1/100), PE-Cy7 CD19 (1/50), PE-Cy5 CD20 (1/50), PE CD3 (5/200), FITC CD3 (2/100), BV786 CD38 (1/50), BUV737 CD27 (1/100).

Donor cells were collected before the start of each organoid culture and stained for the characterization of the HexaPro-specific B cells, as well as B cell and T cell subsets, as described above to determine the baseline of the populations. All data were obtained from FACSymphony™ A5 (BD Biosciences) and analysed on FlowJo™ v10.8.2 software (BD Life Sciences).

Gating strategies for flow cytometry are provided in Supplementary Fig 15.

## SPR measurements of NTD- and RBD-binding antibody fragments

Antibody fragments specific for the NTD and the RBD of SARS-CoV-2 S protein were isolated by phage display technology using in-house libraries based on the IGHV3−23 and IGKV1−39 germline genes displaying Fab or scFv with synthetic diversity introduced into the CDRs, essentially as previously described[58]. Selection was performed on biotinylated antigen SARS-CoV-2 S1 protein (ACRO Biosystems SIN-C82E8) and SARS-CoV-2 RBD protein (ACRO Biosystems SPD-C82E9) caught onto streptavidin-coated paramagnetic beads during 3-4 successive rounds of selection using decreasing amounts of antigen. Enriched fragment-encoding genes were re-cloned into vectors that allowed for production of soluble antibody fragments fused to triple FLAG-tag and a hexahistidine tag[59,60]. Clones encoding antigen-specific binders were identified by ELISA using biotinylated antigen caught on streptavidin and subsequent detection with the horse radish peroxidase-labelled anti-FLAG M2 antibody (Merck F1804). SPR measurements were performed on a Biacore T200 instrument (Cytiva, Sweden) at 25 °C in running buffer HBS-EP (10 mM HEPES pH 7.4, 150 mM NaCl, 3 mM EDTA, 0.05% surfactant P20). By using a capture assay setup, scFv and Fab clones present in bacterial supernatant were captured onto a CM5 series S chip (Cytiva BR1005-30) with an immobilized anti-FLAG M2 antibody (dilution to 20 µg mL⁻¹) or an anti-human KAPPA antibody (Cytiva 28958325, dilution to 20 µg mL⁻¹), respectively. Ancestral S proteins AnSA−5 and AnSA−6, as well as their wt-RBD counterparts and HexaPro, were each injected at 50 nM analyte concentration and were allowed to bind to the captured scFv and Fab clones. Following a dissociation phase, surface regeneration was accomplished by injecting a low pH solution (10 mM glycin-HCl, pH 2.1). Obtained data was subtracted from a reference surface and a blank run using software BIAeval (Cytiva) to retrieve result binding sensorgrams for each clone (Supplementary Fig. 14).

## Reporting summary

Further information on research design is available in the Nature Portfolio Reporting Summary linked to this article.

## Data availability

Maps have been deposited in the Electron Microscopy Data Bank (EMDB) under accession codes EMD-15475 (AnSA-5) and EMD-15482 (AnSA-6). Atomic coordinates have been deposited in the Protein Data

Bank (PDB) under accession codes 8AJA [https://doi.org/10.2210/pdb8AJA/pdb] (AnSA-5) and 8AJL [https://doi.org/10.2210/pdb8AJL/pdb] (AnSA-6). Protein structures used from other publications can be accessed with the pdb accession codes: 6VXX [https://doi.org/10.2210/pdb6VXX/pdb], 6ZOZ [https://doi.org/10.2210/pdb6ZOZ/pdb] and 7BNN [https://doi.org/10.2210/pdb7BNN/pdb]. Source data are provided with this paper for Figs. 3A–D, 4A–C, 5B–D, and 6A–C and Supplementary Figs. 3D–G, 7A–H, 8B, 10, 11, 12, 13A–D, and 14.

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

## Acknowledgements

We thank Anders Olsson for general support, comments and suggestions and the Protein Expression and Characterization Unit at the Science for Life Laboratory for instrument usage. Computational resources were provided by the PDC Center for High Performance Computing at the Royal Institute of Technology (grant number 2019-700, P.-O.S.) and SNIC (SNIC 2021/5-70, P.-O.S.). This work was also supported by the Swedish Foundation of Strategic Research (FFF20-0027, P.-O.S), the European Union's Horizon 2020 research and innovation program (ATAC, 101003650, L.H., H.M., Q.P.-H.), grants from the SciLifeLab National COVID-19 Research Program (J.A.), the Swedish Research Council (2019-01302, 2020-06116, Q.P.-H) and the Knut and Alice Wallenberg Foundation (KAW2020.0102, L.H., Q.P.-H.). Cryo-EM data was collected at the Cryo-EM Swedish National Facility, Stockholm node, funded by the Knut and Alice Wallenberg, Family Erling Persson and Kempe Foundations, SciLifeLab, Stockholm University and Umeå University. The HexaPro construct was a kind gift from Leo Hanke with support from The CoroNAb consortium. A vector encoding a human albumin fused truncated ACE2 was a kind gift from Jan Terje Andersen (University of Oslo, Oslo University Hospital). The following reagent was obtained through BEI Resources, NIAID, NIH: Vector pCAGGS Containing the SARS-CoV-2, Wuhan-Hu-1 Spike Glycoprotein Gene RBD with C-Terminal Hexa-Histidine Tag, NR-52309. Part of this work was performed at the Protein Science Laboratory at Karolinska Institute. The authors thank Magnus Snickars for help with preparation of Fig. 1 and Oriana Ribeiro for the collection and preparation of the tonsils.

## Author contributions

D.H. and K.S. contributed equally. P.-O.S and J.A. designed and supervised research. F.Z. and L.D. designed and performed the plasma sample binding and neutralization experiments. L.H., H.M., and Q.P.-H. contributed to the analysis and interpretation of data. H.P., M.O. and C.H. designed recombinant antibody studies and SPR investigations. K.W. collected and processed the cryo-EM data and D.H. and J.A. built the model. D.H. and K.S. designed and produced the proteins and performed stability experiments. R.S. and S.V. performed tonsil organoid experiments and M.B. provided tonsil samples. D.H., K.S., J.A. and P.-O.S. prepared the manuscript with contributions from all authors. All authors have approved the final version of the manuscript.

## Funding

## Competing interests

D.H., K.S., J.A. and P.-O.S. are co-inventors of the patent application ANCESTRAL PROTEIN SEQUENCES AND PRODUCTION THEREOF with application numbers # 2250542-4 (Swedish Patent Office) and # PCT/SE2023/050423 (PCT), respectively. The other authors declare no competing interests.
