## [Peer Review File · Nature Communications]

Design, structure and plasma binding of ancestral β -CoV scaffold antigensReviewers' Comments:

Reviewer #1:

Remarks to the Author:

Hueting et al. use ancestral sequence reconstruction to generate stabilized spike structures. They characterize two of their constructs biophysically and structurally, showing that they have similar or improved stabilities compared to SARS-CoV-2 S. This is an interesting demonstration of a unique approach to coronavirus vaccine design and the authors present substantial biochemical/biophysical work supporting certain conclusions on antigen stability. My primary critiques are on the evolutionary analysis itself – my overall feeling is that this doesn't necessarily hinder the utility of these constructs as immunological tools, but they are "weird" evolution choices and at the very least mean greater caveats should be given and evolutionary interpretations should be expressly avoided.

Main critique:

A lack of accounting for recombination is a shortcoming that is not adequately addressed from an evolutionary perspective, and the fact that recombination is not incorporated and its implications should be more clearly stated. Lack of consideration of recombination doesn't invalidate these as purely immunological tools, but it does mean that any conclusion about evolution itself is not supported, as these spikes do not represent accurate ancestral sequences. Performing a phylogenetic analysis on a genetic segment that exhibits recombination (which happens for example among NTD, RBD, and S2 domains between lineages, as shown in numerous publications), requires breaking the alignment into non-recombining segments (e.g. as done across the whole genome in PMID 32724171), performing reconstructions on separate segments, and concatenating the matched segments back together at relevant nodes. The lack of accounting of recombination makes it such that any statement about evolution writ large is not well supported (e.g., lines 319-323) and should be removed. Recombination also appears to explain why AnSA-5 doesn't bind to ACE2, a result that is in contrast to other recent publications demonstrating that ancestral RBDs (when properly reconstructed) do bind to ACE2 (PMID 35114688). Specifically, ZC45 and ZXC21 are two viruses in the tree that cluster with SARS-CoV-2 when you build a phylogeny with full spike, but from RBDs they clearly cluster with the clade of viruses including HKU3 which have been shown to evolved a derived loss of ACE2 binding (PMID 35114688) due to presence of two large deletions and many RBD sequence changes. These deletions and the loss of ACE2 binding is a derived trait particular to this clade of RBDs, but you can see in Ext Data Figure 2 alignment that these deletions are inferred in AnSA-5. This occurs because the single phylogeny is built on whole spike without incorporating different evolutionary histories of recombining domains: ZC45 gravitates toward SARS-CoV-2 in the phylogeny despite its RBD being very different, and so the AnSA-5 phylogenetic reconstruction is going to be biased toward reconstructing a non-ACE2-utilizing type of RBD with these deletions because of the imposed paraphyly of ZC45 and the HKU3 group with respect to SARS-CoV-2 despite that not being the true history at the RBD level (PMID 35114688). This likely reduces the immunological efficacy of these spike immunogens, too, because their RBDs resemble the derived clade of HKU3-like spikes which are not the clade thought to pose the biggest spillover threat and have large antigenic changes relative to viruses like SARS-CoV-1 or SARS-CoV-2. However, this last critique depends on the extent to which the authors imagine this being a spike scaffold with use of other RBDs as they show with SARS-CoV-2, or what the intended workflow would be. This is not necessarily a death knell for the entire approach, it is just a large caveat that should be addressed appropriately across multiple parts of the manuscript and implications more clearly stated.

The authors nicely show that these AnSAs do improve certain biophysical traits like spike stability compared to SARS-CoV-2 S. The impact of these conclusions would be higher if it was shown that e.g. immunizing mice with these antigens generated good responses against e.g. SARS-CoV-2 or other bat sarbecoviruses with spillover potential. There is no indication that these proteins (especially if they pack their RBDs down so tightly) would serve as good immunogens, as the experiments that are performed (that SARS-CoV-2-elicited sera can bind to the AnSAs) does not establish this precedent:

the study design is backwards from the question as to whether these AnSAs are good immunogens. (For example, it could be that primarily non-neutralizing antibodies from SARS-CoV-2 sera are the ones exhibiting cross-reactivity, which would not be a good sign.)

Minor critiques:

BANAL viruses from Laos probably not the best example citation of potential for driving future pandemic – since (although not yet demonstrated to my knowledge), there is probably cross-reactivity with SARS-CoV-2 immune sera due to RBD similarity. A useful citation might be for the more divergent Khosta-2 virus which can infect using human ACE2 and has been shown to be antigenically distinct from SARS-CoV-2 (<https://www.biorxiv.org/content/10.1101/2021.12.05.471310v2>)

Why exactly were 9, 6, 5, and 3 chosen for reconstruction? And not e.g. the ancestral sarbecovirus node?

Would be useful in the text to have a broad description of what the nodes of AnSA-5 and -6 (and the others) actually represent – i.e. AnSA-9 is the ancestor of sarbeco- and hibeoviruses, AnSA-6 is seemingly the ancestor of SARS-CoV-2-related spikes and spikes from non-ACE2-utilizing bat sarbecoviruses, and AnSA-5 is seemingly the ancestor of SARS-CoV-2 and SARS-CoV-2-clade bat and pangolin sarbecoviruses (together with ZC45 and ZXC21, which have SARS-CoV-2-clade spike features but HKU3-like RBDs). Related, for Extended Data Figure 1, it might be good to use more “common” names (e.g. isolate) in some of the viruses as opposed to accession numbers, e.g. AVP78031.1 spike protein Bat Sars-like coronavirus is not informative, ZC45 probably means something to more people.

Reviewer #3:

Remarks to the Author:

SARS-CoV-2 spike protein is the major target for the development of vaccines and neutralizing antibodies. Yield and stability are important factors for designing S protein-based vaccines. Based on structural information, S2P and HexaPro were previously designed which greatly improved the yield and stability, especially the HexaPro. In this study, the authors developed a straightforward and low-throughput way to design protein antigens. They used exclusively homologous S sequences to construct ancestral scaffold antigens AnSA-5/6, and obtained high yield even better than that of HexaPro. Cryo-EM structures revealed sole closed conformation with altered hydrogen-bonding network discovered inter-domains and across monomers. Aggregation propensity and thermal stability were tested using DLS and DSF. Finally, AnSA-5/6 wt-RBDs were generated and restored the functions of wt RBD within the AnSAs, indicating that AnSAs are functional and excellent scaffold antigens. The manuscript shows novelty, the experiments are well designed, and most conclusions are convincing. And the scaffold antigens have the potential to be utilized for subunit vaccine development. This reviewer has some experimental remarks and some minor suggestions, hope to help better support the findings and present the data of the study.

Major remarks.

1. In the AnSA-5/6 wt-RBD construct, RBD was exchanged with SARS-CoV-2 wt RBD. These novel RBD-interdomain hydrogen bonds in AnSA-5/6 are likely broken, causing the loss of stability, yielding an increased amount of aggregates. The replacement of RBD domain seems to have some influence on the aggregation propensity and probably also thermal stability. Were the PI-values of AnSA-5/6 wt-RBD tested during storage? And also test the thermal stability using DSF?
2. Did the authors try to solve the structure of AnSA-5/6 wt-RBD or at least do negative staining to confirm these molecules are still intact trimer particles?
3. One of the most important approaches to test the efficiency of a vaccine candidate is animal immunization. Animal protection experiments may not be the scope of this study. It can be helpful to test the total level of IgG and also neutralizing ability of the serum after immunization using the AnSA

wt-RBD antigen.

Minor remarks.

1. Line 515 We furthermore hypothesize that ACE2-binding may be prevented by the fact that the AnSA structures appear to be locked in the closed pre-fusion state (all RBDs down, Fig 2, Extended Data Fig. 5), given that at least one RBD needs to be transiently hinged up to enable ACE2-binding. This hypothesis should be re-considered. Pangolin-CoV spike was also reported to adopt only one conformation with all RBDs down but shows similar affinity to hACE2 compared to that of SARS-CoV-2 spike, which implies that there is not a large energetic cost to opening of the S1 structure. Wrobel, A.G., Benton, D.J., Xu, P. et al. Structure and binding properties of Pangolin-CoV spike glycoprotein inform the evolution of SARS-CoV-2. *Nat Commun* 12, 837 (2021).

The reviewer agrees that the absence of AnSA-ACE2 interaction is attributed to the change in RBD sequence. But it's doubtful AnSA is irreversibly locked in the closed conformation and open conformation can't be induced in presence of its functional receptor.

2. Extended Data Figure 2. It's a little hard to tell which parts of the sequences have more differences without looking at the table under the sequence alignment. This reviewer suggests using labels like * under each residue of the sequence alignment and different background colors to represent the similarity score for each residue (ESPrpt, <http://esprpt.ibcp.fr/ESPrpt/ESPrpt/index.php>). And the name of each domain may be labeled on the top of the sequence alignment.

3. Extended Data Figure 4. For AnSA-5, 959, 111 (37.5%) good particles out of 2,555,520 were used for final 3D reconstruction. For AnSA-6, 253, 429 (12.4%) good particles out of 2,035,826 were used for final 3D reconstruction. AnSA-6 showed much fewer well-organized particles that can be used for 3D reconstruction. The number for HexaPro is ~25.3 %. Does the percentage have any relationship with the stability of the proteins?

4. Extended Data Figure 8 C. Yields not shown of the AnSA-5/6 wt-RBDs.

5. The concentrations of each sample used in the DLS were not given in the method. According to the figure legend of Extended Data Figure 3, different concentrations were used for the four samples, could the authors explain this? Will this affect the results?

6. In general, it's better to have the sample names of either the curves or structures labeled in all figures, not only described in the figure legends.

We have addressed all comments from the editor and reviewers stringently. Our point-by-point responses to comments are given below. All changes introduced in the manuscript are highlighted in yellow. We thank the editor and reviewers for the constructive feedback and their comments that allowed us to further improve our manuscript.

/Per-Olof Syrén on behalf of the authors

Reviewer #1 report:

Huetting et al. use ancestral sequence reconstruction to generate stabilized spike structures. They characterize two of their constructs biophysically and structurally, showing that they have similar or improved stabilities compared to SARS-CoV-2 S. This is an interesting demonstration of a unique approach to coronavirus vaccine design and the authors present substantial biochemical/biophysical work supporting certain conclusions on antigen stability.

We are happy that the referee finds our work interesting.

1. My primary critiques are on the evolutionary analysis itself – my overall feeling is that this doesn't necessarily hinder the utility of these constructs as immunological tools, but they are "weird" evolution choices and at the very least mean greater caveats should be given and evolutionary interpretations should be expressly avoided.

Main

critique:

A lack of accounting for recombination is a shortcoming that is not adequately addressed from an evolutionary perspective, and the fact that recombination is not incorporated and its implications should be more clearly stated.

We thank the referee for these comments and apologize for not being clear on this point in the original version. In the revised version, we have clarified the fact that recombination is not accounted for and added a discussion on implications (on p5, middle and p20 bottom).

2. Lack of consideration of recombination doesn't invalidate these as purely immunological tools, but it does mean that any conclusion about evolution itself is not supported, as these spikes do not represent accurate ancestral sequences. Performing a phylogenetic analysis on a genetic segment that exhibits recombination (which happens for example among NTD, RBD, and S2 domains between lineages, as shown in numerous publications), requires breaking the alignment into non-recombining segments (e.g. as done across the whole genome in PMID 32724171), performing reconstructions on separate segments, and concatenating the matched segments back together at relevant nodes. The lack of accounting of recombination makes it such that any statement about evolution writ large is not well supported (e.g., lines 319-323) and should be removed.

We agree with the referee. The term "ancestral" was kept due to the use of ASR as design method but any explicit statements on coronavirus evolution have been removed. The reference PMID 32724171 was added as reference 20.

3. Recombination also appears to explain why AnSA-5 doesn't bind to ACE2, a result that is in contrast to other recent publications demonstrating that ancestral RBDs

(when properly reconstructed) do bind to ACE2 (PMID 35114688). Specifically, ZC45 and ZXC21 are two viruses in the tree that cluster with SARS-CoV-2 when you build a phylogeny with full spike, but from RBDs they clearly cluster with the clade of viruses including HKU3 which have been shown to have evolved a derived loss of ACE2 binding (PMID 35114688) due to presence of two large deletions and many RBD sequence changes. These deletions and the loss of ACE2 binding is a derived trait particular to this clade of RBDs, but you can see in Ext Data Figure 2 alignment that these deletions are inferred in AnSA-5. This occurs because the single phylogeny is built on whole spike without incorporating different evolutionary histories of recombining domains: ZC45 gravitates toward SARS-CoV-2 in the phylogeny despite its RBD being very different, and so the AnSA-5 phylogenetic reconstruction is going to be biased toward reconstructing a non-ACE2-utilizing type of RBD with these deletions because of the imposed paraphyly of ZC45 and the HKU3 group with respect to SARS-CoV-2 despite that not being the true history at the RBD level (PMID 35114688).

We thank the referee for these comments. We have clarified this point on page 20, bottom in the revised version:

“These sequence differences probably result from the topology of the inferred phylogeny that does not take recombination into account. Specifically, the S protein of Bat SARS-like coronavirus ZC45 shares high overall sequence similarity with SARS-CoV-2 S protein, and therefore localizes to a neighboring branch of SARS-CoV-2 S protein; whereas its RBD is more closely related to non-ACE2-binding S-proteins that differ from SARS-CoV-2 S protein by several point mutations and deletions. These differences are reflected in the reconstructed AnSA sequences and likely interfere with ACE2 binding.”

The reference PMID 35114688 was added as reference 25.

4. This likely reduces the immunological efficacy of these spike immunogens, too, because their RBDs resemble the derived clade of HKU3-like spikes which are not the clade thought to pose the biggest spillover threat and have large antigenic changes relative to viruses like SARS-CoV-1 or SARS-CoV-2. However, this last critique depends on the extent to which the authors imagine this being a spike scaffold with use of other RBDs as they show with SARS-CoV-2, or what the intended workflow would be. This is not necessarily a death knell for the entire approach, it is just a large caveat that should be addressed appropriately across multiple parts of the manuscript and implications more clearly stated.

Please see a response to this comment in point 5 below.

5. The authors nicely show that these AnSAs do improve certain biophysical traits like spike stability compared to SARS-CoV-2 S. The impact of these conclusions would be higher if it was shown that e.g. immunizing mice with these antigens generated good responses against e.g. SARS-CoV-2 or other bat sarbecoviruses with spillover potential. There is no indication that these proteins (especially if they pack their RBDs down so tightly) would serve as good immunogens, as the experiments that are performed (that SARS-CoV-2-elicited sera can bind to the AnSAs) does not establish this precedent: the study design is backwards from the question as to whether these AnSAs are good immunogens. (For example, it could be that primarily non-neutralizing

antibodies from SARS-CoV-2 sera are the ones exhibiting cross-reactivity, which would not be a good sign.)

We thank the referee for this suggestion and have included additional experimental data to address the immunogenicity of AnSAs.

In order to confirm that AnSAs are capable of eliciting an immune response, we have evaluated AnSA-5 immunogenicity using a recently established *in vitro* tonsillar organoid culture model (Wagar *et al.* Nat. Med. PMID: 33432170). The latter has been shown to recapitulate key germinal center human immune responses after immunization with antigens. The added experimental datasets show how *in vitro* immunization with AnSA-5 was able to evoke SARS-CoV-2 S protein specific immune responses in tonsil organoids derived from 3 different donors (with known SARS-CoV-2 infection, COVID-19 vaccination or both). In these experiments, AnSA-5 performed comparably to HexaPro, which was used as a reference. These results are presented in a new section (“AnSA-5 potently boosted immune responses *in vitro*”) which was added on p21. An additional figure 5 (shown below for reference) was added, together with additional extended table and figures. The methods section was updated accordingly.

Figure 5. Adaptive immune responses in tonsil organoids stimulated by AnSA-5 and HexaPro. (A) Workflow for tonsil tissue disruption, culture preparation, antigen immunization, and readouts. **(B)** Levels of HexaPro- and AnSA-5-binding IgG from day 14 organoid cultures. **(C)** Levels of wt, BQ.1.1, and XBB.1.5 RBD-binding IgG from day 14 organoid cultures. The results in the graphs are presented as mean with standard deviation of measurement of duplicates. Data shown as binding antibody unit (BAU) per μg of total IgG detected in tonsil supernatants (BAU/ μg of total IgG) **(D)** HexaPro-specific B cell percentages out of $\text{CD}19^+ \text{CD}20^+$ B cells between BAFF, BAFF and HexaPro, and BAFF and AnSA-5 stimulations from day 14 organoid cultures defined by flow cytometry.

We observed that the *in vitro* immunization of tonsil organoids with AnSA-5 elicited B-cells as well as IgGs that bound HexaPro, which serves as a proxy for the wild-type SARS-CoV-2 S protein (blue bars and dots in Fig. 5B, left and Fig. 5D).

It is highly challenging to obtain samples from donors that are immunogenically naïve towards SARS-CoV-2 S protein due to the progression of the pandemic. However, the induction of anti-AnSA-5 IgGs by AnSA-5 immunization, but not HexaPro immunization in donor T183, illustrates that AnSA-5 is generally capable of not only boosting memory cells but also inducing *de novo* immune responses (blue and orange bars in Fig. 5B, right).

Regarding point No. 4 (above), the new experimental results indicate that immunization with AnSA-5 elicits antibodies that are capable of recognizing the RBD domain of both the wt as well as two currently circulating variants of concern (Omicron BQ.1.1 and XBB.1.5). Finally, it is relevant to note that non-neutralizing antibodies may confer relevant immunogenic protection via complement activation and antibody-mediated phagocytosis, which has been shown to be relevant e.g. for Influenza (PMID: 31914207).

Minor

critiques:

6. BANAL viruses from Laos probably not the best example citation of potential for driving future pandemic – since (although not yet demonstrated to my knowledge), there is probably cross-reactivity with SARS-CoV-2 immune sera due to RBD similarity. A useful citation might be for the more divergent Khosta-2 virus which can infect using human ACE2 and has been shown to be antigenically distinct from SARS-CoV-2 (<https://www.biorxiv.org/content/10.1101/2021.12.05.471310v2>)

We added the suggested reference as reference 11.

7. Why exactly were 9, 6, 5, and 3 chosen for reconstruction? And not e.g. the ancestral sarbecovirus node? Would be useful in the text to have a broad description of what the nodes of AnSA-5 and -6 (and the others) actually represent – i.e. AnSA-9 is the ancestor of sarbeco- and hibecoviruses, AnSA-6 is seemingly the ancestor of SARS-CoV-2-related spikes and spikes from non-ACE2-utilizing bat sarbecoviruses, and AnSA-5 is seemingly the ancestor of SARS-CoV-2 and SARS-CoV-2-clade bat and pangolin sarbecoviruses (together with ZC45 and ZXC21, which have SARS-CoV-2-clade spike features but HKU3-like RBDs).

We thank the reviewer for pointing out that this was not clear in the first version. The corresponding section has been clarified in the revised version (p5, bottom):

“We chose to experimentally study several nodes upstream of SARS-CoV-2 S protein, namely nodes 3, 5, 6 and 9 (Extended Data Fig. 1). Node 3 and 5 were selected because they are closely related to the extant S protein sequence of SARS-CoV-2. Node 6 was chosen as the node closest to SARS-CoV-2 S protein that precedes both the branches of SARS-CoV-2 and a group of sequences closely related to SARS-CoV. Node 9 in turn represents the closest node that precedes S proteins of sarbecoviruses and non-sarbecoviruses, namely hibecoviruses. Therefore, we were also interested in experimentally studying node 9.”

8. Related, for Extended Data Figure 1, it might be good to use more “common” names (e.g. isolate) in some of the viruses as opposed to accession numbers, e.g. AVP78031.1 spike protein Bat Sars-like coronavirus is not informative, ZC45 probably means something to more people.

Extended data figure 1 was extended with common names according to the reviewer’s suggestion. At the same time the accession numbers were kept to allow readers to identify the corresponding database entries.

We thank the referee for the constructive and insightful comments that allowed us to improve our work.

Reviewer #2 (Remarks to the Author):

SARS-CoV-2 spike protein is the major target for the development of vaccines and neutralizing antibodies. Yield and stability are important factors for designing S protein-based vaccines. Based on structural information, S2P and HexaPro were previously designed which greatly improved the yield and stability, especially the HexaPro. In this study, the authors developed a straightforward and low-throughput way to design protein antigens. They used exclusively homologous S sequences to construct ancestral scaffold antigens AnSA-5/6, and obtained high yield even better than that of HexaPro. Cryo-EM structures revealed sole closed conformation with altered hydrogen-bonding network discovered inter-domains and across monomers. Aggregation propensity and thermal stability were tested using DLS and DSF. Finally, AnSA-5/6 wt-RBDs were generated and restored the functions of wt RBD within the AnSAs, indicating that AnSAs are functional and excellent scaffold antigens. The manuscript shows novelty, the experiments are well designed, and most conclusions are convincing. And the scaffold antigens have the potential to be utilized for subunit vaccine development.

We thank the referee for these comments.

This reviewer has some experimental remarks and some minor suggestions, hope to help better support the findings and present the data of the study.

Major remarks.

1. In the AnSA-5/6 wt-RBD construct, RBD was exchanged with SARS-CoV-2 wt RBD. These novel RBD-interdomain hydrogen bonds in AnSA-5/6 are likely broken, causing the loss of stability, yielding an increased amount of aggregates. The replacement of RBD domain seems to have some influence on the aggregation propensity and probably also thermal stability. Were the PI-values of AnSA-5/6 wt-RBD tested during storage? And also test the thermal stability using DSF?

We thank the referee for the comment. AnSA-6 wt-RBD DLS data were included in the previous version of the manuscript. We have now added DLS results for AnSA-5 wt-RBD, showing how wt-RBD variants populate similar peaks compared to HexaPro and AnSA-5/6, respectively. However, the DLS data also illustrate that especially AnSA-6 wt-RBD is less stable and more prone to aggregation than AnSA-6. The results are presented in Extended Data Fig. 12 B and C and are discussed on p26, middle:

“The DLS profile of AnSA-5 harbouring the wt-RBD showed a similar amount of aggregates compared to AnSA-5, whereas the aggregate peak increased for AnSA-6 harbouring the wt-RBD (Extended Data Fig. 12B and C, PI-values 0.60 and 0.83, respectively). The DLS peak present in the mid-range size (ca. 150 nm) that was observed in HexaPro (Extended Data Fig. 3D) is clearly present in the AnSA-6 wt-RBD DLS trace as well.”

PI-values during storage were not recorded for the RBD-exchange variants due to the higher yield of protein required for this experiment.

DSF data were added for both AnSA-5- and AnSA-6- harbouring the wt-RBD (Extended Data, Fig. 12D). The results are discussed on p 27, bottom half:

“The DSF data of AnSA-5/6 harbouring the wt-RBD are largely consistent with the DLS results. AnSA-5 harbouring the wt-RBD seems somewhat stable, with T_m values of 49 and 61 °C (blue line in Extended Data Fig. 12D). The T_m value associated with the first peak (49 °C, RBD domain unfolding) is comparable to both HexaPro (47 °C) and AnSA-5 (50 °C), whereas the T_m value associated with the second peak (61 °C, trimer unfolding) is comparable to HexaPro (63 °C) but decreased compared to AnSA-5 (67 °C). Likewise, AnSA-6 harbouring the wt-RBD was observed to lose some thermostability. This variant exhibits an unclear melting profile with a major melting event between 45 to 65°C (red line in Extended Data Fig. 12D), which may indicate the disintegration of the protein structure, potentially due to shedding of the S1 subunit.”

The purpose of the wt-RBD exchange variants was to show that AnSAs can be utilized as scaffold antigens. In the revised version, we have included new datasets to demonstrate that AnSAs themselves are able to evoke an immune response in pre-immunized tonsil organoids, illuminating their potential as antigens (see response to point 3 below).

2. Did the authors try to solve the structure of AnSA-5/6 wt-RBD or at least do negative staining to confirm these molecules are still intact trimer particles?

As part of the revision process, we have attempted to solve the structures of AnSA-5/6 wt-RBD using the same cryo-EM grids and method as for AnSA-5/6. The AnSA-5/6 structures were solved directly from the first grids frozen and the first dataset collected in a remarkably direct process. In contrast, the AnSA-5/6 wt-RBD are more challenging samples and on the grid; we have not been able to confirm intact trimer particles. This can be due to either the sample itself but could also be due to the grid type used and air-water interface problems, something which is also discussed in Minor comments #3. Resolving this issue would require substantially more grid-optimization and microscope time. The RBD exchange variants are not the main focus of this study and we apologize for not being clear on this point in the original version. We have included new datasets to demonstrate that the AnSAs themselves are able to evoke SARS-CoV-2 S protein specific immune responses in tonsil organoids derived from 3 different donors (with known SARS-CoV-2 infection, COVID-19 vaccination or both; see response to point 3 below).

3. One of the most important approaches to test the efficiency of a vaccine candidate is animal immunization. Animal protection experiments may not be the scope of this study. It can be helpful to test the total level of IgG and also neutralizing ability of the serum after immunization using the AnSA wt-RBD antigen.

We thank the referee for this comment.

In order to confirm that AnSAs are capable of eliciting an immune response, we have evaluated AnSA-5 immunogenicity using a recently established *in vitro* tonsillar organoid culture model (Wagar *et al.* Nat. Med. PMID: 33432170). The latter has been shown to recapitulate key germinal center human immune responses after immunization with antigens. The added experimental datasets show how *in vitro* immunization with AnSA-5 was able to evoke SARS-CoV-2 S protein specific immune responses in tonsil organoids derived from 3 different donors (with known SARS-CoV-2 infection, COVID-19 vaccination or both). In these experiments, AnSA-5 performed comparably to HexaPro, which was used as a reference. These results are presented in a new section (“AnSA-5 potentially boosted immune responses

in vitro) which was added on p21. An additional figure 5 (shown below for reference) was added, together with additional extended table and figures. The methods section was updated accordingly.

Figure 5. Adaptive immune responses in tonsil organoids stimulated by AnSA-5 and HexaPro. (A) Workflow for tonsil tissue disruption, culture preparation, antigen immunization, and readouts. **(B)** Levels of HexaPro- and AnSA-5-binding IgG from day 14 organoid cultures. **(C)** Levels of wt, BQ.1.1, and XBB.1.5 RBD-binding IgG from day 14 organoid cultures. The results in the graphs are presented as mean with standard deviation of measurement of duplicates. Data shown as binding antibody unit (BAU) per μg of total IgG detected in tonsil supernatants (BAU/ μg of total IgG). **(D)** HexaPro-specific B cell percentages out of $\text{CD}19^+ \text{CD}20^+$ B cells between BAFF, BAFF and HexaPro, and BAFF and AnSA-5 stimulations from day 14 organoid cultures defined by flow cytometry.

We observed that the *in vitro* immunization of tonsil organoids with AnSA-5 elicited B-cells as well as IgGs that bound HexaPro, which serves as a proxy for the wild-type SARS-CoV-2 S protein (blue bars and dots in Fig. 5B, left and Fig. 5D).

It is highly challenging to obtain samples from donors that are immunogenically naïve towards SARS-CoV-2 S protein due to the progression of the pandemic. However, the induction of anti-AnSA-5 IgGs by AnSA-5 immunization, but not HexaPro immunization in donor T183, illustrates that AnSA-5 is generally capable of not only boosting memory cells but also inducing *de novo* immune responses (blue and orange bars in Fig. 5B, right).

Minor remarks.
 1. Line 515 We furthermore hypothesize that ACE2-binding may be prevented by the fact that the AnSA structures appear to be locked in the closed pre-fusion state (all RBDs down, Fig 2, Extended Data Fig. 5), given that at least one RBD needs to be transiently hinged up to enable

ACE2-binding.

This hypothesis should be re-considered. Pangolin-CoV spike was also reported to adopt only one conformation with all RBDs down but shows similar affinity to hACE2 compared to that of SARS-CoV-2 spike, which implies that there is not a large energetic cost to opening of the S1 structure. Wrobel, A.G., Benton, D.J., Xu, P. et al. Structure and binding properties of Pangolin-CoV spike glycoprotein inform the evolution of SARS-CoV-2. *Nat Commun* 12, 837 (2021).

The reviewer agrees that the absence of AnSA-ACE2 interaction is attributed to the change in RBD sequence. But it's doubtful AnSA is irreversibly locked in the closed conformation and open conformation can't be induced in presence of its functional receptor.

We thank the referee for making us aware of this reference and agree regarding the revising of the hypothesis. We have clarified this point in the revised version on p20, bottom by focusing on sequence differences as a result of the tree topology.

2. Extended Data Figure 2. It's a little hard to tell which parts of the sequences have more differences without looking at the table under the sequence alignment. This reviewer suggests using labels like * under each residue of the sequence alignment and different background colors to represent the similarity score for each residue (ESPrpt, <http://esprpt.ibcp.fr/ESPrpt/ESPrpt/index.php>). And the name of each domain may be labeled on the top of the sequence alignment.

We thank the referee for the suggestions and have revised Extended data Figure 2 accordingly.

3. Extended Data Figure 4. For AnSA-5, 959, 111 (37.5%) good particles out of 2,555,520 were used for final 3D reconstruction. For AnSA-6, 253, 429 (12.4%) good particles out of 2,035,826 were used for final 3D reconstruction. AnSA-6 showed much fewer well-organized particles that can be used for 3D reconstruction. The number for HexaPro is ~25.3 %. Does the percentage have any relationship with the stability of the proteins?

The number of good particles that can be used for the final reconstruction can be indicative of protein stability. However, it is not a reliable measure to use since any disintegration of the complex observed on the grid could also be due to the grid type used as well as effects of the air-water interface on the cryo-EM grid upon freezing. The effect on the protein sample by the air-water interface can be unrelated to the stability of the protein complex in solution, which is why we are relying on other techniques to measure protein stability.

4. Extended Data Figure 8 C. Yields not shown of the AnSA-5/6 wt-RBDs.

We thank the referee for being observant. Yields were added in Extended Data Figure 12A.

5. The concentrations of each sample used in the DLS were not given in the method. According to the figure legend of Extended Data Figure 3, different concentrations were used for the four samples, could the authors explain this? Will this affect the results?

As indicated in the figure legend of Extended Data Fig. 3, most samples were measured at a concentration of 1.2 - 1.5 mg/mL. We performed control experiments in which the concentration of a reference sample (HexaPro) was analyzed at 1.5 mg/mL and 0.7 mg/mL,

showing that the ratio of aggregate to soluble peak was not strongly affected by this change in concentration (see below, figure not included in manuscript).

The reason for using a lower concentration with the sample AnSA-3 (Extended Data Fig. 3E) is because this variant expresses at very low quantities (see also SDS-PAGE in panel A of the same figure), which is why we had lower yields of this variant available for this experiment. As mentioned in the manuscript, these properties make AnSA-3 a less suitable antigen, which is why we did not focus on this variant.

6. In general, it's better to have the sample names of either the curves or structures labeled in all figures, not only described in the figure legends.

The reason for formatting the figures this way is because the instructions for submitting to Nature specified to not include legends in the figures ("Where possible, text, including keys to symbols, should be provided in the legend rather than on the figure itself"). If the editor deems this suggested change necessary, we will update the figures at the next stage.

We thank the referee for their helpful and insightful suggestions and comments that allowed us to improve our work.

Editorial comments,

POLICIES AND FORMS REQUIRED FOR RESUBMISSION

* Please complete or update the following checklist(s) to verify compliance with our research ethics and data reporting standards. Address all points on the checklist, revising your manuscript in response to the points if needed. The form(s) must be downloaded and completed in Adobe Reader rather than opened in a web browser. Each form must be uploaded as a Related Manuscript file at the time of resubmission.

Editorial policy checklist:

<https://www.nature.com/documents/nr-editorial-policy-checklist.pdf>

Reporting summary:

Revised versions of the editorial policy checklist and reporting summary were provided.

DATA AND CODE AVAILABILITY

* All Nature Communications manuscripts must include a “Data Availability” section after the Methods section but before the References. If any of the data can only be shared on request or are subject to restrictions, please specify the reasons and explain how, when, and by whom the data can be accessed. For more information on this policy and a list of examples, see: <https://www.nature.com/documents/nr-data-availability-statements-data-citations.pdf>

The data availability statement is available after the Methods section before additional Method references.

* To maximise the reproducibility of research data, we strongly encourage you to provide a file containing the raw data underlying the following types of display items:

- Any reported means/averages in box plots, bar charts, and tables
- Dot plots/scatter plots, especially when there are overlapping points
- Line graphs

The data should be provided in a single Excel file with data for each figure/table in a separate sheet, or in multiple labelled files within a zipped folder. Name this file or folder ‘Source Data’, and include a brief description in your cover letter. The “Data Availability” section should also include the statement “Source data are provided with this paper.”

To learn more about our motivation behind this policy, please see:

<https://www.nature.com/articles/s41467-018-06012-8>

An excel file was provided and the cover letter was updated.

We thank the editor for the constructive comments that helped us to improve our work.

Reviewers' Comments:

Reviewer #1:

Remarks to the Author:

My original critiques were appropriately addressed in revision.

Reviewer #3:

Remarks to the Author:

My questions have been addressed. Especially, they used an in vitro tonsillar organoid culture model to prove that AnSA-5 is able to elicit a good immune response. The antibodies could even recognize WT RBD and also RBD from two VOCs, omicron BQ.1.1 and XBB.1.5. It would be interesting to see if the AnSA-5 elicited antibodies could neutralize WT SARS-CoV-2, omicron BQ.1.1 and XBB.1.5 or not.

Reviewer #4:

Remarks to the Author:

In their analysis of both the AnSA-5 and HexaPro antigens using human tonsil organoids the authors present data showing that AnSA-5 Ag produce antibody, B and Tfh responses to SARS-CoV-2 RBDs including variants like BQ.1.1 and XBB.1.5 that are comparable to a HexaPro antigen. The use of tonsil organoids for this purpose is admirable given the implications arising from such data for vaccine design. However, there are a number of problems with the data presented stemming from the lack of a proper characterisation of the tonsil organoids through to the immunological analyses performed.

The organoids appear to be relatively incompletely characterised. The authors have only shown they have clusters of tonsillar cells growing in transwells but there is no definition of the many cellular markers to truly determine that the cellular clusters have developed polarity and formed organoids with functional germinal centers. The authors should better characterise their organoids by using confocal microscopy of cultures at various stages from the initial plating through to day 15. Without this information, it is difficult to be certain that cultures have developed into organoids.

The cultures also appear to have been grown in the presence of BAFF alone while other growth factors that may be critical for the development of tonsil organoids (as with many other kinds of organoids) have not been explored or included. Have the authors tried to grow organoids in the presence of HGF, noggin, PGE2, nicotinamide, human FGF10 and bFGF all of which appear to be essential for the development of functional tonsil organoids.

The optimal conditions required to grow and characterise human tonsil organoids have recently been reported in detail by Kim et al.,(Biomaterials 283 (2022) 121460; <https://doi.org/10.1016/j.biomaterials.2022.121460>). The authors may find this paper useful to review and revise their methodologies.

As a baseline, the authors should show that the organoids have developed from EpCAM positive cells, consistent with an epithelial origin and that organoids are EpCAM+. It is also important to show the expression of CD44, ITGA6 and NGFR as evidence of differentiation and maturation into tonsil organoids and of cellular proliferation (Ki67). Did the authors embed fresh dissociated tonsil cells into matrigel or were the cells recovered from frozen harvests?

Confocal microscopy would also help to show that NGFR, ITGA6, or CD44 are localized on the outer side of mature organoids, while markers like MUC1 should be present in the interior of tonsil organoids. This distribution of markers would provide a level of confidence that organoids with the correct architecture have been successfully grown. Further confirmation of organoid architecture

should be possible using histological analyses with haematoxylin and eosin, Alcian blue, periodic acid-Schiff and Masson's trichrome.

Having successfully cultured structured tonsil organoids, it is also important to show that the organoids express ACE2 (on the outer side), TMPRSS2 and furin. Once the tonsil organoids have been fully characterised, interpreting the comprehensive immunological analyses could follow.

Additional points:

Figure 5 is very confusing. There are no statistical analyses provided for any of the comparisons. By presenting the data as BAU per μg of total IgG it is very difficult to gauge the strength of Ab responses. This is particularly true of the BQ.1.1 and XBB.1.5 RBD responses.

The antibody responses against BQ.1.1 and XBB.1.5, presented as BAU/ μg IgG in Figure 5 appear relatively weak compared with the wild type RBD. What are the GMTs?

Perhaps a better way to determine the breadth of anti-RBD reactivity is to perform Multiplex-RBD assays. This would greatly strengthen this section of analyses rather than relying on limited RBD based ELISAs.

The authors also claim that the use of the tonsil organoids is a more relevant way to look at immune responses for humans. They show the development of anti-RBD responses but don't show that these antibodies are neutralising in an sVNT assay or by TCID50 against cell culture derived SARS-CoV-2 variants. This is important to show and should be provided.

In 5D, why is only the HexPro specific B cell response shown? Was there also an AnSA-5 specific B cell response?

Were any analyses performed to look at affinity maturation, isotype switching and somatic hypermutation (eg BCR repertoires, IGVH gene usage)? These would be important as part of the analyses of the antigen (AnSA-5) specific B cell responses.

Figure 10. Although increasing the overall number of graphs, the results of the B and T cell composition would be easier to understand if the data was presented as separate bar graphs rather than tile plots that make the separation of frequency of responses difficult to appreciate. In addition, the tile plots do not show error bars or levels of significance for the various comparisons.

The ELISA results in figure 11 show relatively weak or non-existent anti-AnSA-5 responses and don't support the comment that AnSA-5 is able to produce antibody responses that are of a comparable capacity to HexaPro. With low Ab responses it may be better to present the data in OD plots or Ab titres rather than as BAU/ μg IgG.

We have addressed all comments from the editor and reviewers stringently. Our point-by-point responses to comments are given below. All changes introduced in the manuscript are highlighted in yellow. We thank the editor and reviewers for the constructive feedback and their comments that allowed us to further improve our manuscript.

/Per-Olof Syrén on behalf of the authors

REVIEWER COMMENTS

Reviewer #1 (Remarks to the Author)

My original critiques were appropriately addressed in revision.

We are happy to see that the referee is satisfied with our revision.

Reviewer #3 (Remarks to the Author):

My questions have been addressed. Especially, they used an in vitro tonsillar organoid culture model to prove that AnSA-5 is able to elicit a good immune response. The antibodies could even recognize WT RBD and also RBD from two VOCs, omicron BQ.1.1 and XBB.1.5. It would be interesting to see if the AnSA-5 elicited antibodies could neutralize WT SARS-CoV-2, Omicron BQ.1.1 and XBB.1.5 or not.

In the revised version of the manuscript, a pseudovirus neutralization assay was included. Because of the limited amount of supernatant and low antibody concentration, we conducted the pseudovirus neutralization assay solely against the SARS-CoV-2 wt pseudovirus. The results were added in Extended Data Fig. 12 and in the results section p. 26, line 634-640.

“We subsequently measured the neutralization activity against the wt SARS-CoV-2 pseudovirus in the culture supernatant of immunized tonsil organoids. We detected neutralization activity for donor T147 immunized with AnSA-5 (50% neutralization titre (NT50): 14.05) and HexaPro (NT50: 232.5) as well as donor T182 immunized with AnSA-5 (NT50: 7.29) (Extended Data Fig. 12). For T183, the low titre of anti-RBD antibodies, as measured in ELISA (Fig. 5C, left panel), were not sufficient to detect neutralization in the pseudovirus assay. These results thus suggest that the AnSA-5 protein can induce neutralizing antibodies against the wt spike.”

Extended Data Figure 12. Neutralization activity against wt pseudovirus in organoid cultures.

Two-fold dilutions of day 14 organoid culture supernatants (starting from 1:5 dilution) were tested. Mean \pm standard deviation of duplicates for one representative experiment is shown for organoids from donors T147 and T182.

The protocol for the pseudovirus neutralization assay was added in Material and methods on p. 41, line 1133-1157, as described below:

“Pseudovirus neutralization assay

The gene encoding the wild-type S protein lacking the C-terminal 19 codons ($S_{\Delta 19}$) was synthesized by GenScript using human codon optimization. To generate (HIV-1/NanoLuc2AEGFP)-SARS-CoV-2 particles, three plasmids were used, with NanoLuc luciferase and EGFP reporter genes (pCCNanoLuc2AEGFP), HIV-1 structural/regulatory proteins (pHIV_{NL}GagPol) and SARS-CoV-2 $S_{\Delta 19}$ carried by separate plasmids, as previously described.^{56,57} 293FT cells were transfected with 7 μ g of pHIV_{NL}GagPol, 7 μ g of pCCNanoLuc2AEGFP, and 2.5 μ g of pSARS-CoV-2- $S_{\Delta 19}$ carrying the $S_{\Delta 19}$ gene from the G614 strain of SARS-CoV-2 (at a molar plasmid ratio of 1:1:0.45) using 66 μ l of 1 mg mL⁻¹ PEI.

Twofold serial dilutions of tonsil organoid culture supernatants (1:5 - 1:640) were incubated with pseudovirus carrying the wt S protein from SARS-CoV-2 for 1 h at 37 °C. The mixture was subsequently incubated with 293T-hACE2 cells for 48 h, after which the cells were washed with PBS and lysed with Luciferase Cell Culture Lysis reagent (Promega). NanoLuc luciferase activity in the lysates was measured using the Nano-Glo Luciferase Assay System (Promega) with a Tecan Infinite microplate reader. The relative luminescence units were normalized to those derived from cells infected with pseudotyped virus in the absence of tonsil organoid culture supernatant. The neutralization titre 50 (NT₅₀) was expressed as the maximal dilution of the culture supernatant where the reduction of the signal is $\geq 50\%$. The experiment was performed in duplicates and the mean neutralization (%) values are presented in Extended Data Fig. 12. The NT₅₀ values were determined using four-parameter nonlinear regression (the least squares regression method without weighting) (GraphPad Prism 7.04 software). As a positive control, fivefold dilutions of the monoclonal IgG anti-RBD antibody SA55 ranging from 1 μ g mL⁻¹ to 0.01 ng mL⁻¹ were used.⁵⁸

We thank the referee for the constructive and insightful comments that allowed us to improve our work.

Reviewer #4 (Remarks to the Author):

In their analysis of both the AnSA-5 and HexaPro antigens using human tonsil organoids the authors present data showing that AnSA-5 Ag produce antibody, B and Tfh responses to SARS-CoV-2 RBDs including variants like BQ.1.1 and XBB.1.5 that are comparable to a HexaPro antigen. The use of tonsil organoids for this purpose is admirable given the implications arising from such data for vaccine design. However, there are a number of problems with the data presented stemming from the lack of a proper characterisation of the tonsil organoids through to the immunological analyses performed.

The organoids appear to be relatively incompletely characterised. The authors have only shown they have clusters of tonsillar cells growing in transwells but there is no definition of the many cellular markers to truly determine that the cellular clusters have developed polarity and formed organoids with functional germinal centers. Without this information, it is difficult to be certain that cultures have developed into organoids.

The cultures also appear to have been grown in the presence of BAFF alone while other growth factors that may be critical for the development of tonsil organoids (as with many other kinds of organoids) have not been explored or included. Have the authors tried to grow organoids in the presence of HGF, noggin, PGE2, nicotinamide, human FGF10 and bFGF all of which appear to be essential for the development of functional tonsil organoids.

The optimal conditions required to grow and characterise human tonsil organoids have recently been reported in detail by Kim et al., (Biomaterials 283 (2022) 121460; <https://doi.org/10.1016/j.biomaterials.2022.121460>). The authors may find this paper useful to review and revise their methodologies.

The authors would like to thank the referee for the comments and remarks.

Organoids are commonly three-dimensional structures derived from stem cells or tissue samples that can mimic the structure and function of specific organs. However, unless immune cells are deliberately introduced, organoids typically do not naturally contain immune cells. Likewise, the organoid model described in the paper recommended by Reviewer #4 recapitulates the key characteristics of the tonsil epithelium but does not contain immune cells. After enzymatic digestion of the tonsil tissues, the EpCAM positive cells were flow-sorted and subsequently cultured in matrigels to generate human tonsil epithelial organoids. The basal epithelial cells express key molecules required for SARS-CoV-2 entry, including angiotensin-converting enzyme 2 (ACE2), transmembrane serine protease 2 (TMPRSS2), and furin. Consequently, these cells are susceptible to infection by SARS-CoV-2.

There are thus significant differences in the human tonsil epithelial organoid model utilized in Kim et al.'s study and the immune organoid model developed by Wagar et al. (Nat Med 2021, PMID: 33432170), which was employed in the present study. The immune organoid model, which is derived from tonsil immune cells, does not include epithelial cells. The model is particularly relevant for studying the immune response to specific protein antigens, including influenza, other viral vaccines, and pathogens (Wagar et al. Nat Med, PMID: 33432170). The immune organoid recapitulates key features of an adaptive immune response, encompassing the activation and differentiation of T and B cells, expansion of antigen-specific T and B cells,

somatic hypermutation, class switching, and the production of specific antibodies (Wagar et al. 2021).

In our study, we followed the sample preparation method described in Wagar et al. 2021. Immune cells, either mixed or not with HexaPro or AnSA-5 antigens, were used for the generation of the organoids.

The procedure was illustrated in Fig. 5A and described in Material and methods, p. 40, line 1100-1107 with reference to Wagar et al:

“Tonsil organoids were then prepared following the sample preparation method as described by Wagar et al.³⁶ Specifically, tonsil tissues were dissected and cell suspensions were prepared, enumerated, and resuspended to 2×10^7 cells mL^{-1} .³⁶ Fresh cells were plated, 100 μL per well, into permeable (0.4- μm pore size) membranes (48-well size PTFE or polycarbonate membranes in standard 24-well plates) with the lower chamber consisting of complete medium (1 mL) supplemented with $0.5 \mu\text{g mL}^{-1}$ of recombinant human B cell-activating factor (BAFF; BioLegend #559608) as a control or $0.5 \mu\text{g mL}^{-1}$ BAFF and either AnSA-5 or HexaPro at a final concentration of $0.15 \mu\text{g mL}^{-1}$. Cultures were incubated at 37 $^{\circ}\text{C}$, 5% CO_2 with humidity.”

The tonsil organoid cultures were maintained in the presence of recombinant human B-cell activating factor (BAFF) as this was found to be the minimum requirement in our experiments to sustain good viability of the cell cultures (see Fig. below). The growth factors suggested by the reviewer are typically used for stem cell cultures and epithelial cell differentiation, and therefore, they are not relevant for our model.

Percentage of live cells in the organoid culture at day 7 and day 14, termed cells, in the presence or not of BAFF. Baseline values represent the HexaPro-specific B cell percentages at the beginning of the cultures; colors represent the different donors.

Using this immune organoid model, we showed that immunization with AnSA-5 resulted in both production of Abs against VOC, and HexaPro-specific B cells, further supporting the hypothesis that AnSA-5 can serve as a scaffold to elicit immune responses for future SARS-CoV-2 variants.

In the revised version, we have explicitly stated in the abstract, introduction and the results parts that the tonsil organoid model employed in the present study is an immune organoid model to distinguish it from conventional organoid model derived from stem cells, as described below.

Abstract, p. 2, line 53

“...in an immune organoid model derived from tonsils...”

Introduction, p. 4, line 116-118

“We show how an AnSA is able to induce or boost a specific immune response in an immune organoid model derived from tonsils of infected and/or vaccinated donors (referred to as “tonsil organoids” from here on).”

Results, p21, line 545-547:

“The tonsil organoid model employed in the present study is an immune organoid model³⁶, different from conventional organoid models³⁷ derived from stem cells.”

The citation to Kim et al. was added as reference 37.

As a baseline, the authors should show that the organoids have developed from EpCAM positive cells, consistent with an epithelial origin and that organoids are EpCAM+. It is also important to show the expression of CD44, ITGA6 and NGFR as evidence of differentiation and maturation into tonsil organoids and of cellular proliferation (Ki67). Did the authors embed fresh dissociated tonsil cells into matrigel or were the cells recovered from frozen harvests?

Confocal microscopy would also help to show that NGFR, ITGA6, or CD44 are localized on the outer side of mature organoids, while markers like MUC1 should be present in the interior of tonsil organoids. This distribution of markers would provide a level of confidence that organoids with the correct architecture have been successfully grown. Further confirmation of organoid architecture should be possible using histological analyses with haematoxylin and eosin, Alcian blue, periodic acid–Schiff and Masson’s trichrome.

Having successfully cultured structured tonsil organoids, it is also important to show that the organoids express ACE2 (on the outer side), TMPRSS2 and furin. Once the tonsil organoids have been fully characterised, interpreting the comprehensive immunological analyses could follow.

As previously mentioned, there are significant differences between the models developed by Kim et al. and Wagar et al. These differences imply that each model can be employed for distinct applications and purposes. In contrast to the model developed by Kim et al., our model does not rely on EpCAM positive cells or matrigel and is not intended as an infection model specifically for SARS-CoV-2. As described earlier, fresh single-cell suspensions from tonsil tissues were plated into the wells of permeable membrane plates (commonly known as Transwells) along with the antigen of interest. The immune tonsil organoid model is designed to replicate the germinal centre reactions and is utilized for studying the adaptive immune response to antigens. Due to the absence or minimal presence of epithelial cells in the model,

it is not suitable for studying the expression of epithelial cell markers such as GFR, ITGA6, or CD44, or the proteins ACE2, TMPRSS2, or furin, which are involved in SARS-CoV-2 infection.

Microscopic analysis of GCs in the organoid system can provide valuable insights into the cellular localization and polarization within the GC. Wagar et al. analyzed the embedded, frozen sections of organoids of tonsil donors by fluorescent confocal microscopy on day 4 and found evidence of light and dark zone organization, as indicated by the separation of CD83 and CXCR4 staining B cells, respectively, which are characteristic of GCs.

However, it should be noted that this method alone does not serve as an absolute standard for analyzing antigen-specific responses. Other studies have demonstrated that flow cytometry and ELISA data can provide sufficient information in this regard (Schaefer-Babajew Nature 2023, PMID: 36473496, Dan et al. Science 2021, PMID: 33408181). In our study, flow cytometric analysis of the organoid model showed five distinct B cell phenotypes, i.e. germinal centre (GC), pre-GC, memory, plasmablasts (PBs), and Naïve B cells (Extended Data Fig. 10, p. 24). Resolution of the above cellular populations with characteristic surface markers provide direct evidence for the existence of germinal centres, as previously reported by Wagar et al. The stronger decrease in the pre-GC pool of B cells from day 7 to 14 in the antigen-stimulated groups compared to the respective BAFF control groups (Extended Data Fig. 10A, middle and right, p. 24) for these two donors indicates an antigen-induced B cell response. Furthermore, the functionality of these GCs is proven by indirectly measuring the humoral response, through specific Abs against a variety of SARS-CoV-2 variants, and the existence of HexaPro-specific B cells produced upon immunization with HexaPro or AnSA-5 (Fig. 5, p. 21). Finally, the HexaPro-specific B cells residing within all functional B cell phenotypes provides further direct evidence for the functionality of the GCs in our model (Extended Data Fig. 10, p. 24).

Additional points:

Figure 5 is very confusing. There are no statistical analyses provided for any of the comparisons. By presenting the data as BAU per μg of total IgG it is very difficult to gauge the strength of Ab responses. This is particularly true of the BQ.1.1 and XBB.1.5 RBD responses.

The antibody responses against BQ.1.1 and XBB.1.5, presented as BAU/ μg IgG in Figure 5 appear relatively weak compared with the wild type RBD. What are the GMTs?

As the tonsil donors had distinct histories of vaccination and infection, we presented the results of antibody titers and HexaPro-specific B cells for each organoid individually. Each column in the graph represents the antibody titers, measured in duplicates, for each organoid and condition.

Across all three organoids, the antibody titers against RBD and HexaPro were 7- to 428-fold higher in the organoids immunized with HexaPro or AnSA-5 compared to the BAFF control. Additionally, the anti-RBD titers were found to be higher in all three organoids that were immunized with HexaPro compared to AnSA-5. As the data is presented for each organoid individually, and due to the limited amount of organoids tested, we are unable to include statistical analysis on the graph.

In order to better appreciate the increase in antibody level and HexaPro-specific B cells in stimulated cells compared to the BAFF-control, we now present the results differently in Fig. 5B-D (p. 21) and modified the figure legend accordingly:

Figure 5. Adaptive immune responses in tonsil organoids stimulated by AnSA-5 and HexaPro.

(A) Workflow for tonsil tissue disruption, culture preparation, antigen immunization, and functional readouts. **(B)** Levels of HexaPro- and AnSA-5-binding IgG from day 14 organoid cultures (T147, T182 and T183) stimulated with HexaPro or AnSA-5 compared to BAFF control. **(C)** Levels of wt, BQ.1.1, and XBB.1.5 RBD-binding IgG from day 14 organoid cultures stimulated with HexaPro or AnSA-5 compared to BAFF control. In **(B)** and **(C)**, the antibody titres in the graphs are presented as mean (represented by dots) of measurement of duplicates in one experiment. Data shown as binding antibody unit (BAU) per μg of total IgG detected in tonsil supernatants (BAU μg^{-1} of total IgG). **(D)** HexaPro-specific B cell percentages out of CD19⁺ CD20⁺ B cells between BAFF, BAFF and HexaPro, and BAFF and AnSA-5 stimulations from day 14 organoid cultures defined by flow cytometry. Colours represent different donors.

The following sentence was changed (p. 24, line 614-616):

“The titres of anti-HexaPro antibodies elicited by HexaPro and AnSA-5 were of the same magnitude (Fig. 5B), highlighting that the two antigens have a comparable capacity of inducing S-protein binding antibodies.”

Please also see our response to the reviewer's comments on Extended Data Fig. 11 below, regarding our choice of BAU per μg of total IgG for the presented data.

Perhaps a better way to determine the breadth of anti-RBD reactivity is to perform Multiplex-RBD assays. This would greatly strengthen this section of analyses rather than relying on limited RBD based ELISAs.

The main objective of this part of the study was to evaluate whether the AnSA protein could stimulate a human immune response. Our results demonstrated that AnSA-5 was able to elicit both a specific antibody response and B cell response against HexaPro (Fig. 5B and D, p. 21), indicating its potential as a vaccine antigen. In the future, we plan to investigate binding to RBD from other variants. To this end, we will also consider incorporating Multiplex-RBD assays.

The authors also claim that the use of the tonsil organoids is a more relevant way to look at immune responses for humans. They show the development of anti-RBD responses but don't show that these antibodies are neutralising in an sVNT assay or by TCID50 against cell culture derived SARS-CoV-2 variants. This is important to show and should be provided.

In the revised version of the manuscript, a pseudovirus neutralization assay was included. Because of the limited amount of supernatant and low antibody concentration, we conducted the pseudovirus neutralization assay solely against the SARS-CoV-2 wt pseudovirus. The results were added in Extended Data Fig. 12 and in the results section p. 26, line 634-640.

"We subsequently measured the neutralization activity against the wt SARS-CoV-2 pseudovirus in the culture supernatant of immunized tonsil organoids. We detected neutralization activity for donor T147 immunized with AnSA-5 (50% neutralization titre (NT50): 14.05) and HexaPro (NT50: 232.5) as well as donor T182 immunized with AnSA-5 (NT50: 7.29) (Extended Data Fig. 12). For T183, the low titre of anti-RBD antibodies, as measured in ELISA (Fig. 5C, left panel), were not sufficient to detect neutralization in the pseudovirus assay. These results thus suggest that the AnSA-5 protein can induce neutralizing antibodies against the wt spike."

Extended Data Figure 12. Neutralization activity against wt pseudovirus in organoid cultures.

Two-fold dilutions of day 14 organoid culture supernatants (starting from 1:5 dilution) were tested. Mean \pm standard deviation of duplicates for one representative experiment is shown for organoids from donors T147 and T182.

The protocol for the pseudovirus neutralization assay was added in Material and methods on p. 41, line 1133-1157, as described below:

“Pseudovirus neutralization assay

The gene encoding the wild-type S protein lacking the C-terminal 19 codons ($S_{\Delta 19}$) was synthesized by GenScript using human codon optimization. To generate (HIV-1/NanoLuc2AEGFP)-SARS-CoV-2 particles, three plasmids were used, with NanoLuc luciferase and EGFP reporter genes (pCCNanoLuc2AEGFP), HIV-1 structural/regulatory proteins (pHIV_{NL}GagPol) and SARS-CoV-2 $S_{\Delta 19}$ carried by separate plasmids, as previously described.^{56,57} 293FT cells were transfected with 7 μ g of pHIV_{NL}GagPol, 7 μ g of pCCNanoLuc2AEGFP, and 2.5 μ g of pSARS-CoV-2- $S_{\Delta 19}$ carrying the $S_{\Delta 19}$ gene from the G614 strain of SARS-CoV-2 (at a molar plasmid ratio of 1:1:0.45) using 66 μ l of 1 mg mL⁻¹ PEI.

Twofold serial dilutions of tonsil organoid culture supernatants (1:5 - 1:640) were incubated with pseudovirus carrying the wt S protein from SARS-CoV-2 for 1 h at 37 °C. The mixture was subsequently incubated with 293T-hACE2 cells for 48 h, after which the cells were washed with PBS and lysed with Luciferase Cell Culture Lysis reagent (Promega). NanoLuc luciferase activity in the lysates was measured using the Nano-Glo Luciferase Assay System (Promega) with a Tecan Infinite microplate reader. The relative luminescence units were normalized to those derived from cells infected with pseudotyped virus in the absence of tonsil organoid culture supernatant. The neutralization titre 50 (NT₅₀) was expressed as the maximal dilution of the culture supernatant where the reduction of the signal is $\geq 50\%$. The experiment was performed in duplicates and the mean neutralization (%) values are presented in Extended Data Fig. 12. The NT₅₀ values were determined using four-parameter nonlinear regression (the least squares regression method without weighting) (GraphPad Prism 7.04 software). As a positive control, fivefold dilutions of the monoclonal IgG anti-RBD antibody SA55 ranging from 1 μ g mL⁻¹ to 0.01 ng mL⁻¹ were used.⁵⁸

In 5D, why is only the HexaPro specific B cell response shown? Was there also an AnSA-5 specific B?

The primary objective of this experiment was to assess the potential of the ancestral protein in stimulating an immune response against SARS-CoV-2. Our findings demonstrated that immunization with AnSA-5 resulted in the generation of HexaPro-specific B cells in organoids obtained from all three donors (Fig. 5D, p. 21). These results provide further evidence supporting the hypothesis that AnSA-5 can serve as a scaffold for eliciting immune responses

and hold promise as a potential booster vaccine candidate to combat future emerging SARS-CoV-2 variants. Additionally, the presence of antibodies against AnSA-5 in ELISA following immunization suggests the induction of AnSA-5-specific B cells (Fig. 5B). This is illustrated in particular by the observation that anti-AnSA-5 antibodies were induced by immunization with AnSA-5, but not HexaPro in donor T183.

Were any analyses performed to look at affinity maturation, isotype switching and somatic hypermutation (eg BCR repertoires, IGVH gene usage)? These would be important as part of the analyses of the antigen (AnSA-5) specific B cell responses.

Previous data has already illustrated that antigen-driven somatic hypermutation, affinity maturation and class switching are supported in the immune organoid derived from tonsils used in our study (Wagar et al. 2021). The primary focus of our study was to evaluate whether the AnSA protein can elicit an antibody response against the SARS-CoV-2 spike, we thus believe that the current data (specific antibody response and B and T cell profiling) is sufficient to support the functionality of the GCs in the organoid model and our conclusion on the AnSA protein (can elicit an antibody response against SARS-CoV-2 spike protein). We acknowledge that obtaining information about somatic hypermutation, isotype switching, and affinity maturation statuses would be valuable, and will be considered in future experiments.

Figure 10. Although increasing the overall number of graphs, the results of the B and T cell composition would be easier to understand if the data was presented as separate bar graphs rather than tile plots that make the separation of frequency of responses difficult to appreciate. In addition, the tile plots do not show error bars or levels of significance for the various comparisons.

Since the data is presented for each organoid individually and the number of organoids tested is limited, we are unable to include statistical analysis on the graph. To provide a clearer visualization of the frequency of the B cell subset, we have now represented the percentages as a dot plot in Extended Data Fig. 10 (p. 24).

Extended Data Figure 10. Cell composition of B and T cell types in organoid cultures. (A) Percentages of different B cell populations out of CD19⁺CD20⁺ B cells at days 7 and 14 of the organoid cultures. Error bars indicate mean with the SD of two to three independent measurements of each population with flow cytometry. **(B)** Percentages of different B cell populations out of HexaPro-specific B cells at day 14 of the organoid cultures. **(C)** Percentages of different T cell populations out of CD19⁻CD3⁺ T cells at days 7 and 14 of the organoid cultures. Baseline values presented in the plots were measured at the beginning of the cultures; HexaPro and AnSA-5 stimulated organoids include BAFF.

The ELISA results in figure 11 show relatively weak or non-existent anti-AnSA-5 responses and don't support the comment that AnSA-5 is able to produce antibody responses that are of a comparable capacity to HexaPro. With low Ab responses it may be better to present the data in OD plots or Ab titers rather than as BAU/ μ g IgG.

The primary objective was to demonstrate that AnSA-5 could elicit antibodies against the spike (HexaPro) and RBD from different SARS-CoV-2 variants. We have shown that the antibody response against HexaPro and RBD from the wild type (wt), BQ.1.1, and XBB.1.5 was comparable following stimulation with either HexaPro or AnSA-5 (Fig. 5b, c, p. 21 and Extended Fig. 11, right panel, p. 25). Given that the most potent neutralizing antibodies typically target the RBD region, our results suggest that the AnSA-5 protein has the capability

to induce antibodies against the functional portion of the protein. This observation confirms that AnSA-5 has the capacity to induce a broad-spectrum antibody response.

Expression in BAU/ml using the WHO international standard is recommended to study the immune response to SARS-CoV-2. The calibrated serum with a known concentration in BAU/ml can be used to generate the standard curve, and different dilutions of the supernatant from the BAFF control (1/10) and immunized organoid (1/10 and 1/50) can be tested.

Since the amount of total IgG increased over the 14-days period and varied between conditions (Figure A below), we normalized the antibody level per amount of total IgG for comparison (Extended data Fig. 11, p. 25). In the picture B and C below, specific antibodies measured by ELISA are expressed as OD₄₅₀ nm using a 1/10 dilution of the organoid culture supernatant. However, the signal for anti-HexaPro, AnSA-5 and RBD is saturated for some organoids immunized with AnSA-5 or HexaPro at this dilution, which is one of the reasons we prefer to use another dilution for those samples (1/50) and normalize with the calibrated WHO serum. Consequently, we opt to present the data as BAU/μg of total IgG in the manuscript (Fig. 5 and Extended Fig. 11) to account for these variations.

Total IgG (μg/ml) (A), HexaPro- and AnSA-5-binding IgG (OD₄₅₀ nm) (B), and anti-RBD IgG antibodies (OD₄₅₀ nm) (C) from day 14 organoid cultures (T147, T182 and T183) stimulated with HexaPro or AnSA-5 compared to BAFF control. In B and C, the organoid culture supernatants were diluted 1/10.

We thank the referee for the constructive and insightful comments that allowed us to improve our work.

Reviewers' Comments:

Reviewer #4:

Remarks to the Author:

The authors have now responses to the comments raised in my review.

1. Characterisation of organoids. The explanation and clarification provided by the authors is satisfactory. Whilst further characterisation of the 'immune organoids' might be of interest, this was not the primary objective of the current paper and the authors have shown that the organoids have produced a meaningful immune response - NAb responses.

2. Antibody responses: the presentation of data in Figure 5 is now clearer and easier to follow.

3. NT data: The presentation of NAb using pseudovirus neutralisation assays adds significantly to the quality of the overall findings. The low Ab titres and limited supernatant volumes limited the breadth of sVNT assays performed however, the results presented in supp Fig 12 supports the production of functional Ab responses.

4. BCR repertoires IGVH usage: the main focus of the study was achieved and reinforced with the new data included without these additional studies.

5. Figure 10: the presentation of the data dot plots makes the separation of the B and T cell responses in different donor samples much clearer.

The remainder of the questions and comments have also been adequately addressed.

We have addressed all comments stringently. Our point-by-point responses to the referee's comments are given below. All changes introduced in the manuscript are high-lighted in yellow. We thank the editor and reviewers for the constructive feedback and their comments that allowed us to further improve our manuscript.

/Per-Olof Syrén on behalf of the authors

REVIEWER COMMENTS

Reviewer #4 (Remarks to the Author):

The authors have now responses to the comments raised in my review.

1. Characterisation of organoids. The explanation and clarification provided by the authors is satisfactory. Whilst further characterisation of the 'immune organoids' might be of interest, this was not the primary objective of the current paper and the authors have shown that the organoids have produced a meaningful immune response - NAb responses.
2. Antibody responses: the presentation of data in Figure 5 is now clearer and easier to follow.
3. NT data: The presentation of NAb using pseudovirus neutralisation assays adds significantly to the quality of the overall findings. The low Ab titres and limited supernatant volumes limited the breadth of sVNT assays performed however, the results presented in supp Fig 12 supports the production of functional Ab responses.
4. BCR repertoires IGVH usage: the main focus of the study was achieved and reinforced with the new data included without these additional studies.
5. Figure 10: the presentation of the data dot plots makes the separation of the B and T cell responses in different donor samples much clearer.

The remainder of the questions and comments have also been adequately addressed.

We are happy that the referee is satisfied with our revision. We thank the referee for the constructive and insightful comments